nature
microbiology
## OPEN

# Large freshwater phages with the potential to augment aerobic methane oxidation

Lin-Xing Chen [1], Raphaël Méheust [1], Alexander Crits-Christoph[2], Katherine D. McMahon[3], Tara Colenbrander Nelson[4], Gregory F. Slater[5], Lesley A. Warren[4,5] and Jillian F. Banfield [1,2,6,7] ✉

There is growing evidence that phages with unusually large genomes are common across various microbiomes, but little is known about their genetic inventories or potential ecosystem impacts. In the present study, we reconstructed large phage genomes from freshwater lakes known to contain bacteria that oxidize methane. Of manually curated genomes, 22 (18 are complete), ranging from 159 kilobase (kb) to 527 kb in length, were found to encode the *pmoC* gene, an enzymatically critical subunit of the particulate methane monooxygenase, the predominant methane oxidation catalyst in nature. The phage-associated PmoC sequences show high similarity to (>90%), and affiliate phylogenetically with, those of coexisting bacterial methanotrophs, including members of *Methyloparacoccus*, *Methylocystis* and *Methylobacter* spp. In addition, pmoC-phage abundance patterns correlate with those of the coexisting bacterial methanotrophs, supporting host–phage relationships. Future work is needed to determine whether phage-associated PmoC has similar functions to additional copies of PmoC encoded in bacterial genomes, thus contributing to growth on methane. Transcriptomics data from Lake Rotsee (Switzerland) showed that some phage-associated *pmoC* genes were highly expressed in situ and, of interest, that the most rapidly growing methanotroph was infected by three pmoC-phages. Thus, augmentation of bacterial methane oxidation by pmoC-phages during infection could modulate the efflux of this potent greenhouse gas into the environment.

Bacteriophages (phages), viruses that infect and replicate within bacteria, are important in both natural and human microbiomes because they prey on bacterial hosts, mediate horizontal gene transfer, alter host metabolism and redistribute bacterially derived compounds via host cell lysis[1]. A phenomenon that has recently come to light via metagenomic studies is the prominence of phages with genomes that are much larger than the average size of ~55 kilobases (kb) predicted based on current genome databases[2]. The recently reported phage genomes range up to 735 kb in length and encode a diversity of genes involved in transcription and translation, as well as genes that may augment host metabolism[2]. Augmentation of bacterial energy generation by auxiliary metabolic genes has been reported for phages with smaller genomes. For example, some encode photosynthesis-related enzymes[3,4], some deep-sea phages have sulfur oxidation genes[5] and others that infect marine ammonia-oxidizing Thaumarchaeota harbour a homologue of ammonia monooxygenase subunit C (that is, *amoC*)[6,7]. Unreported to date is the role of phages involved in the oxidation of methane, a greenhouse gas that is 20–23 times more effective than $CO_2$ (ref. [8]). Biological oxidation of methane is largely driven by microorganisms, including aerobic methanotrophs belonging to Alphaproteobacteria, Gammaproteobacteria and Verrucomicrobia[9,10] which use soluble methane monooxygenases (sMMOs) and/or particulate methane monooxygenases (pMMOs)[11]. The pMMO, the predominant methane oxidation catalyst in nature, is a 300-kDa trimeric metalloenzyme[12] that converts methane to methanol in the periplasm[11,13]. It is encoded by the *pmoCAB* operon[14] and some bacterial genomes encode

multiple *pmoCAB* operons as well as additional copies of *pmoC* that appear to be essential for growth on methane[15].

We considered the possibility that phages infecting methanotrophs could directly impact methane oxidation and thus methane release. Phages with very large genomes were recently reported from a man-made lake that covers a deposit of methane-generating tailings from an oil sands mine in Canada[2]. In the present study, we searched the unreported phage genomic fragments from this lake for genes involved in methane oxidation. We identified four assembled fragments that encoded the enzymatically critical *pmoC* subunit of pMMOs. Hereafter, we refer to these phages as pmoC-phages. We also investigated the metagenomic datasets from freshwater lakes Crystal Bog and Lake Mendota in Madison, WI, United States and Lake Rotsee in Switzerland[16], which are known sources of sediment-derived methane[17], and found examples of *pmoC* on a subset of phage genomic fragments from all three ecosystems. Of the 22 pmoC-phage genomes, 18 were manually curated to completion, enabling verification that they do not encode the *pmoA* or *pmoB* subunit of pMMOs. All complete and partial genomes are >159 kb in length. Microbial communities from all three lakes are known to contain proteobacterial methanotrophs[18,19], some of which we infer are the hosts of the pmoC-phages. We suggest that pmoC-phages may play important roles in the methane cycle when infecting their bacterial hosts.

## Results

**Active methane oxidation in a methane-generating tailings lake.** Oil sands (bituminous sands) deposits are mined for petroleum and generate large volumes of waste that produce methane, hydrogen

[1]Department of Earth and Planetary Sciences, University of California, Berkeley, CA, USA. [2]Department of Plant and Microbial Biology, University of California, Berkeley, CA, USA. [3]Departments of Civil and Environmental Engineering, and Bacteriology, University of Wisconsin, Madison, WI, USA. [4]Department of Civil and Mineral Engineering, University of Toronto, Toronto, Ontario, Canada. [5]School of Geography and Earth Science, McMaster University, Hamilton, Ontario, Canada. [6]Department of Environmental Science, Policy, and Management, University of California, Berkeley, CA, USA. [7]Earth and Environmental Sciences, Lawrence Berkeley National Laboratory, Berkeley, CA, USA. ✉e-mail: jbanfield@berkeley.edu

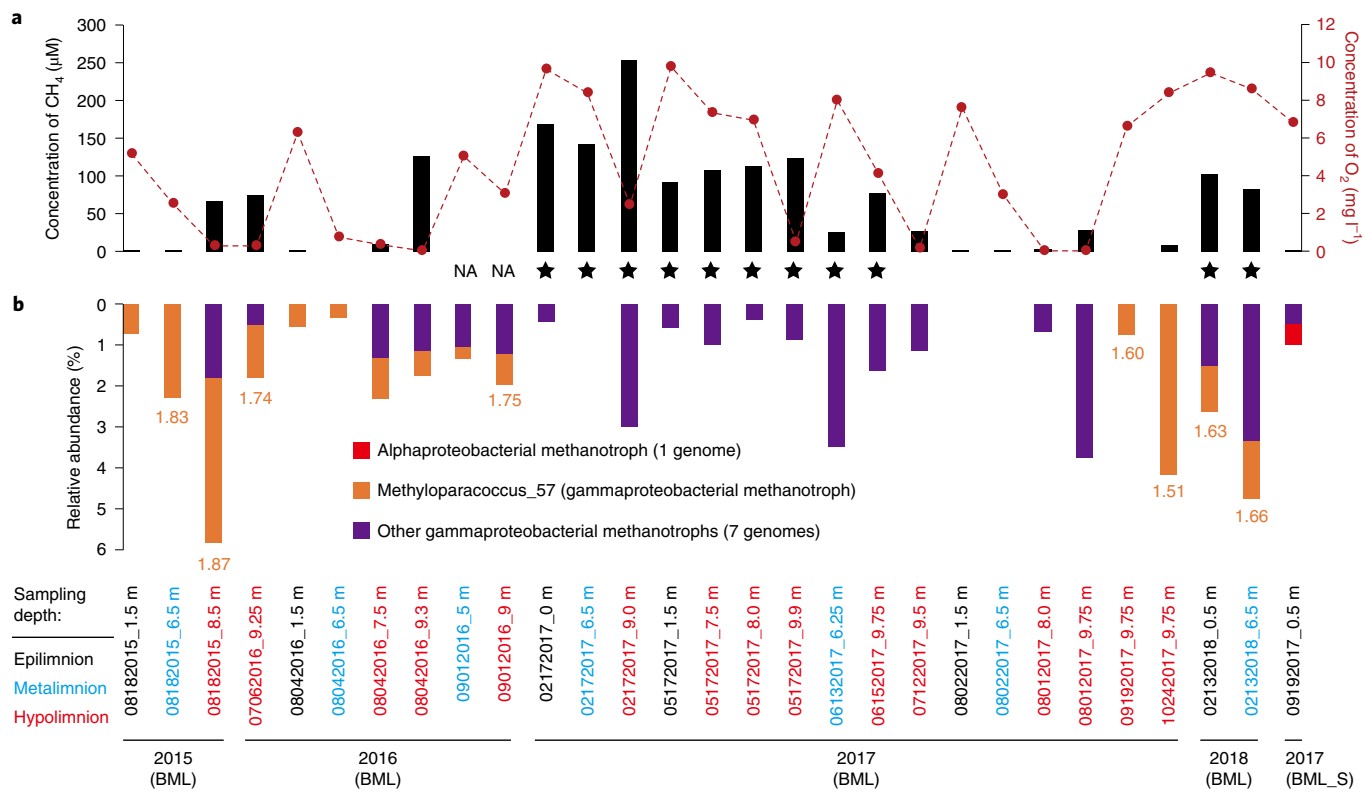

**Fig. 1 | Geochemical and biological evidence for methane oxidation in BML and BML_S samples. a**, The methane and oxygen concentrations at different depths at each sampling time point. Samples in which methanotrophs are inferred to be less active or inactive are indicated by stars. NA, not available. **b**, The relative abundances of methanotrophs. The iRep values (orange font) indicative of the growth rates of Methyloparacoccus_57 are shown; values for other methanotrophs are provided in Supplementary Fig. 2.

sulfide and ammonia[20]. The oil sands reclamation pit lake of Base Mine Lake (BML) in Alberta (Canada) was constructed by placing a layer of water over a tailings deposit, with the long-term goal of developing a lake ecosystem supported by a stable water-cap oxic zone, which would permit the oxidation of methane, hydrogen sulfide and ammonia. The Base Mine Lake is characterized by high concentrations of dissolved methane and ammonia (up to 253 μM and 73.5 μM, respectively; Fig. 1a and see also Supplementary Table 1), especially in the hypolimnetic zone (the lower layer of water in a thermally stratified lake) and at the tailings–water interface, reflecting mobilization of these reductants from the underlying tailings[21,22]. Stable isotope analysis (d[13]C, d[2]H) of pore water methane from within the tailings indicated that the methane was produced via fermentation by indigenous methanogenic archaea[23]. We observed a notable sink of methane in the hypolimnion (Fig. 1a and see also Supplementary Table 1). For example, in 2015 and 2016, oxygen was driven to almost undetectable levels (<5 μM) and dissolved methane decreased rapidly, moving up into the water cap from the tailings–water interface, suggesting that the indigenous bacterial communities used methane as a primary carbon source for growth.

Genome-resolved metagenomics was used to identify microorganisms involved in methane oxidation in the 28 BML water samples and 1 sample from the freshwater source of the lake (BML source, BML_S) (Supplementary Fig. 1, Supplementary Dataset 1). We reconstructed genomes of eight gammaproteobacterial methanotrophs that were collectively more abundant in the hypolimnion than in the upper layers (Student's $t$-test, $P = 0.0190$), and one alphaproteobacterial methanotroph from BML_S (Fig. 1b, and see also Supplementary Figs. 2–4 and Supplementary Table 2). Genes encoding sMMOs and/or pMMOs were detected in these genomes,

and some contain more than one copy of the *pmoCAB* operon and also stand-alone *pmoC* (see Supplementary Figs. 5 and 6, and Supplementary Tables 2 and 3). The most frequently detected methanotroph in BML, Methyloparacoccus_57 (see Supplementary Fig. 7), shares 96.3% 16S ribosomal RNA (rRNA) gene sequence similarity with that of *Methyloparacoccus murrellii* strain R-49797 (ref. [24]). Methyloparacoccus_57 may be a key player in methane oxidation because it had a higher growth rate (iRep values of 1.51–1.87) than any other aerobic methanotrophs (iRep values of 1.32–1.61) coexisting in the communities, especially in the hypolimnion of 2015 and 2016 (Fig. 1 and see also Supplementary Fig. 2). Methane accumulated in lake samples collected from February to June 2017 and in February 2018 despite the availability of oxygen (Fig. 1a and see also Supplementary Table 1), suggesting that low temperatures probably slowed down the activity of the methanotrophs. Reanalysis of published metagenomic datasets of oil sands from Canada[25–28] detected Methyloparacoccus_57 in other sites (see Methods and Extended Data Fig. 1), suggesting their potentially significant role in the sink of methane in such systems.

**Phages with stand-alone *pmoC* genes.** Genomes of huge phages from the BML samples have been previously reported[2], but many other phage genomic fragments remain to be analysed. We searched the full set of phage fragments for genes that could contribute to methane oxidation and found *pmoC* genes that shared >86% amino-acid identity with those of published bacterial methanotrophs (see Supplementary Table 4). Many of the phage scaffolds ended at or near the *pmoC* gene (see Supplementary Table 4), apparently because the assembly was confounded by very similar *pmoC* genes encoded in coexisting bacterial genomes. Manual scaffold extension confirmed no gene encoding *pmoA/pmoB* located

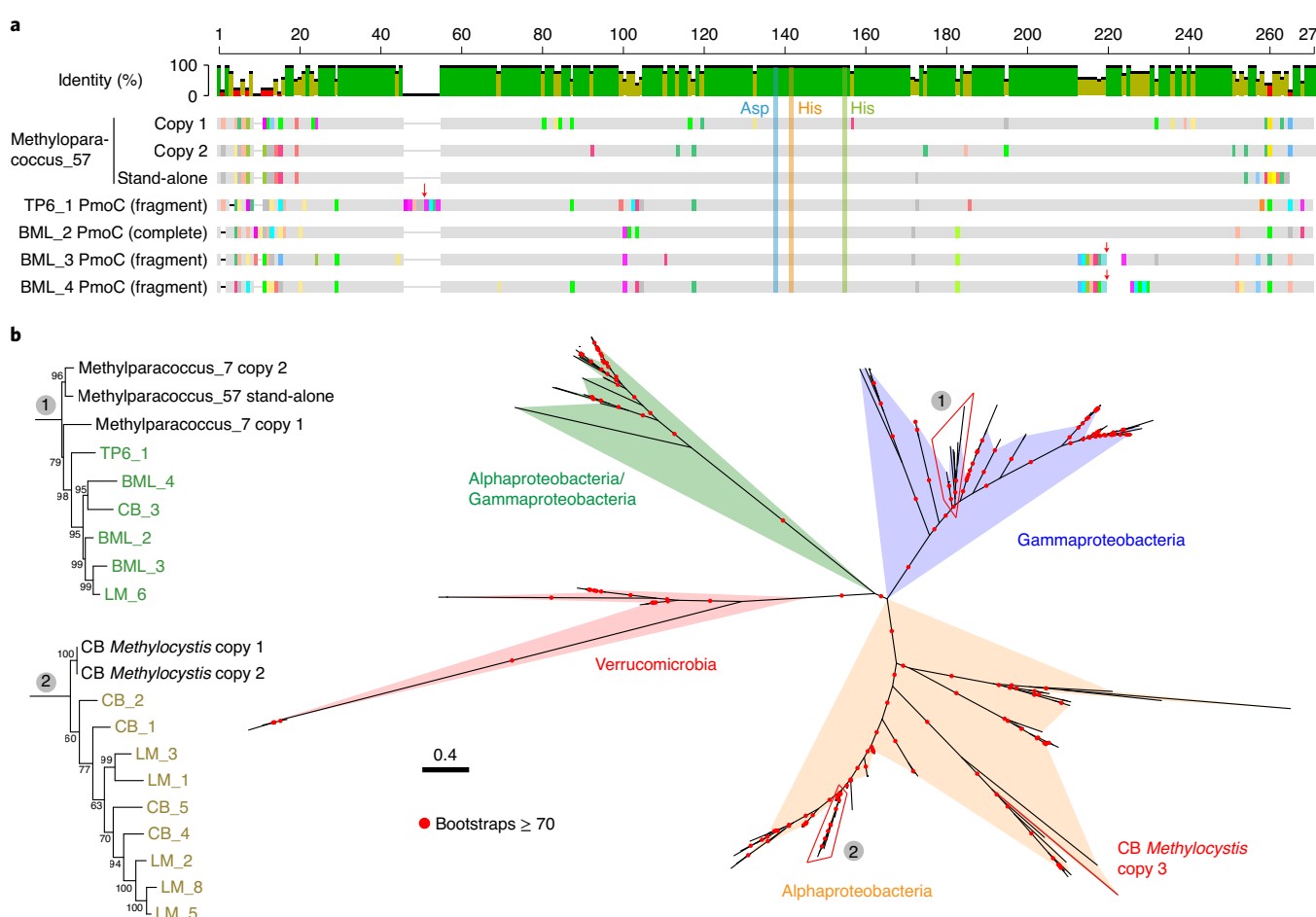

**Fig. 2 | Bacterial and phage-associated PmoC. a**, Alignment of some bacterial and phage-associated PmoC sequences. The three residues in PmoC for copper ion coordination are highlighted. The *pmoC* genes of TP6_1, BML_3 and BML_4 are fragmented (red arrows) and both pieces are shown (see Supplementary Fig. 10 for full alignment). **b**, Phylogenetic analysis of bacterial and phage-associated PmoC. The coloured regions show the clades of published and currently reported bacterial sequences. The phylogenies of phage-associated PmoC are shown in detail. The CB *Methylocystis* sp. has a stand-alone copy of *pmoC* (CB *Methylocystis* copy 3).

nearby (see Supplementary Figs. 8 and 9, for example). One of these pmoC-phage genomes from BML samples (that is, BML_4) was curated to completion (circularized; see 'Genomic features and taxonomy of pmoC-phages' section) to confirm the absence of *pmoA*/*pmoB* in the genome.

Reanalysis of the published oil sands datasets detected one pmoC-phage scaffold (TP6_1) in a Suncor tailings pond sample collected in 2012 (see Methods)[27]. In addition, phages similar to TP6_1 and BML_3 were detected in two other samples from Alberta (see Extended Data Fig. 2), that is, TP_MLSB collected in 2011 (ref. [27]) and PDSYNTPWS collected in 2012 (ref. [26]). From PDSYNTPWS, we curated a phage genome without *pmoC* (referred to as 'PDSYNTPWS_1'), which is 99% similar to BML_3 (64% and 75% of genomes aligned, respectively). Our reanalysis of published ¹³CH₄-based DNA-SIP (stable isotope probing) data[26] detected PDSYNTPWS_1 in the heavy DNA-SIP fraction (see Extended Data Fig. 3). This fraction was dominated by Methyloparacoccus_57. Based on the co-occurrence of the host and phage in a sample in which biological methane oxidation was demonstrated isotopically, we suggest that Methyloparacoccus_57 may have been the host for phage PDSYNPWS_1. Also supporting this association is the high genomic and phylogenetic similarity between PDSYNPWS_1 and BML_3 (Fig. 3), the host for which was predicted as Methyloparacoccus_57.

To test for phage-associated *pmoC* in other lakes reported to emit methane[17], we searched our previously published metagenomic datasets from Lake Mendota (LM) and Crystal Bog (CB) in Madison, WI, United States[19], and those recently published from Lake Rotsee (LR) in Switzerland[16]. The LM, CB and LR datasets were reanalysed (see Methods), and Hidden Markov Model (HMM)-based searches detected *pmoC* on phage scaffolds from all the three lakes (see Supplementary Table 5), suggesting the potentially wide distribution of related phages in habitats with methane.

We confirmed the high similarity of the bacterial and phage-associated PmoC predicted from all datasets to PmoC of previously described alphaproteobacterial and gammaproteobacterial methanotrophs (see Supplementary Tables 4 and 5). Alignment of these PmoC sequences with references from well-known bacterial methanotrophs[29] confirmed the presence of the residues necessary for the copper-binding site, that is, Asp 156, His 160 and His 173 (Fig. 2a and see also Supplementary Fig. 10) and required for O₂ binding and methane oxidation[12,30]. It is of interest that the bacterial and phage-associated PmoC sequences were generally very similar in the central membrane- and periplasma-associated portions, but divergent at the cytoplasmic N and C termini. The *pmoC* genes in four of the pmoC-phages were fragmented into two pieces and another one (LM_8) contained only the C terminus (see Supplementary Fig. 11). In addition, the *pmoC* gene from CB_5

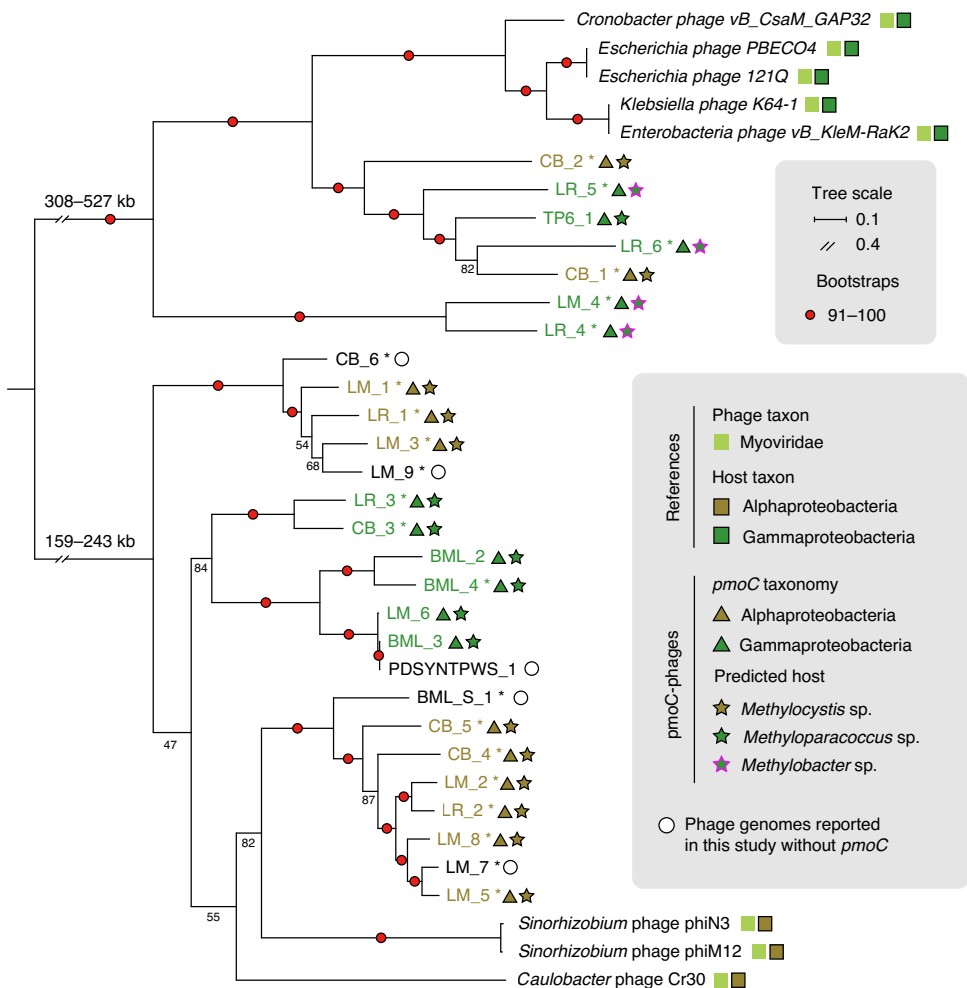

**Fig. 3 | Phylogeny and predicted host of pmoC-phages.** The complete phage genomes reported here are indicated by asterisks. The genome size ranges of the two groups of phages are shown. The taxonomy of phages and their hosts are indicated by coloured squares, triangles or stars. The bootstrap values are indicated by red circles when ≥91 or shown as numbers. See Supplementary Fig. 19 for phylogeny based on DNA polymerase sequences.

exhibited within-population variation, because a subset of phages lacked the central region where the active site is located.

Regardless of sampling sites, phage-associated PmoC often clustered phylogenetically, although sequences from phages with Alphaproteobacteria versus Betaproteobacteria hosts clustered separately (Fig. 2b and see also Supplementary Fig. 12). Moreover, the phage-associated PmoC was always more similar (>90%) to the PmoC of bacterial methanotrophs coexisting in the communities than to the published bacterial PmoC. It is interesting that the total abundance of phage-associated *pmoC* was higher than that of bacterial *pmoC* in some samples (see Supplementary Fig. 13).

**Genomic features and taxonomy of pmoC-phages.** A total of 22 unique pmoC-phage scaffolds with sequencing coverage ≥20× were selected for manual curation to completion and 18 were completed (no gaps and circular; Table 1 and see also Supplementary Tables 4 and 5). In addition, one partial and four complete genomes of closely related phages, but without *pmoC*, were manually reconstructed for comparison (see 'Metabolic potentials of pmoC-phages and their relatives' section). The phage genomes are 159–527 kb in length (GC content: 32–44%), and encode between 224 and 594 ORFs and up to 29 transfer RNAs (tRNA; Table 1). The phage tRNAs correspond with the most commonly used codons in the phage genomes (see Supplementary Fig. 14).

To measure the intrapopulation heterogeneity of pmoC-phages, we identified single nucleotide polymorphisms (SNPs) in BML_2, the most frequently detected pmoC-phage in BML samples. The BML_2 population was highly clonal and displayed little genetic diversity across different depths and sampling time points (see Supplementary Fig. 15 and also Supplementary Information).

Notably, PDSYNTPWS_1 and BML_3, which were sampled from the same region of Canada but in different years, share high genomic similarity with LM_6 (from Lake Mendota), but differ in the *pmoC* region (see Supplementary Fig. 16). PDSYNTPWS_1 does not contain the *pmoC* gene or the five neighbouring genes found in BML_3, and LM_6 has *pmoC* (not fragmented) but lacks the five neighbouring genes. It is interesting that LM_1, LM_7 and LM_8 from Lake Mendota share a 2-kb region near the partial *pmoC* of LM_8. This region encodes hypothetical, phage-associated and bacterial genes, including part of an acyl-coenzyme A (CoA) dehydrogenase (see Supplementary Fig. 17) and may be present due to recombination that occurred during coinfection. The similarity of acyl-CoA dehydrogenase to a gene from *Methylocystis* spp. may indicate that this bacterium is the host (see 'Predicted hosts of pmoC-phages' section).

Eight published complete phage genomes (155–358 kb in length) related to those reported here were retrieved based on ViPTree analyses (see Supplementary Fig. 18)[31,32] and included in protein family

**Table 1 | General features of the manually curated phage genomes**

| Sampling site | Sampling year | Genome name (short name) | Length (bp) | GC content (%) | No. of ORFs | No. of tRNAs | Complete or partial | *pmoC* taxonomy |
|---|---|---|---|---|---|---|---|---|
| BML (Canada) | 2015–2017 | BML_pmoC-phage_2 (BML_2) | 218,687 | 33 | 342 | 15 | Partial | Gamma- |
| | | BML_pmoC-phage_3 (BML_3)[b] | 190,971 | 34 | 272 | 20 | Partial | Gamma- [c,d] |
| | | BML_pmoC-phage_4 (BML_4) | 243,619 | 34 | 342 | 18 | Complete | Gamma-[c] |
| BML_S (Canada) | 2017 | BML_S_phage_1 (BML_S_1) | 167,437 | 40 | 212 | 24 | Complete | – |
| TP6 (Canada) | 2012 | TP6_pmoC-phage_1 (TP6_1) | 308,538 | 37 | 406 | 29 | Partial | Gamma-[c] |
| PDSYNTPWS (Canada) | 2012 | PDSYNTPWS_phage_1 (PDSYNTPWS_1)[b] | 222,435 | 34 | 358 | 20 | Partial | – |
| Lake Mendota (Madison, WI, United States) | 2008–2012 | Lake_Mendota_pmoC-phage_1 (LM_1) | 174,291 | 41 | 249 | 21 | Complete | Alpha- |
| | | Lake_Mendota_pmoC-phage_2 (LM_2) | 174,276 | 39 | 263 | 24 | Complete | Alpha- |
| | | Lake_Mendota_pmoC-phage_3 (LM_3) | 172,382 | 41 | 249 | 21 | Complete | Alpha- |
| | | Lake_Mendota_pmoC-phage_4 (LM_4) | 353,177 | 32 | 465 | 14 | Complete | Gamma- |
| | | Lake_Mendota_pmoC-phage_5 (LM_5) | 166,198 | 38 | 245 | 19 | Complete | Alpha- |
| | | Lake_Mendota_pmoC-phage_6 (LM_6)[b] | 198,907 | 34 | 313 | 20 | Partial | Gamma- |
| | | Lake_Mendota_phage_7 (LM_7) | 166,826 | 39 | 238 | 25 | Complete | – |
| | | Lake_Mendota_pmoC-phage_8 (LM_8) | 167,952 | 39 | 252 | 25 | Complete | Alpha-[e] |
| | | Lake_Mendota_phage_9 (LM_9) | 172,107 | 40 | 240 | 22 | Complete | – |
| Crystal Bog (Madison, WI, United States) | 2007–2009 | Crystal_Bog_pmoC-phage_1 (CB_1) | 352,383 | 35 | 445 | 23 | Complete | Alpha- |
| | | Crystal_Bog_pmoC-phage_2 (CB_2) | 527,138 | 38 | 594 | 13 | Complete | Alpha- |
| | | Crystal_Bog_pmoC-phage_3 (CB_3) | 166,456 | 35 | 247 | 18 | Complete | Gamma- |
| | | Crystal_Bog_pmoC-phage_4 (CB_4) | 165,508 | 38 | 264 | 4 | Complete | Alpha- |
| | | Crystal_Bog_pmoC-phage_5 (CB_5) | 166,149 | 44 | 248 | 4 | Complete | Alpha-[d] |
| | | Crystal_Bog_phage_6 (CB_6) | 174,375 | 38 | 233 | 0 | Complete | – |
| Lake Rotsee (Switzerland) | 2017–2018 | Lake_Rotsee_pmoC-phage_1 (LR_1) | 168,581 | 40 | 224 | 4 | Complete | Alpha- |
| | | Lake_Rotsee_pmoC-phage_1 (LR_2) | 160,734 | 37 | 241 | 6 | Complete | Alpha- |
| | | Lake_Rotsee_pmoC-phage_1 (LR_3) | 159,173 | 35 | 248 | 10 | Complete | Gamma- |
| | | Lake_Rotsee_pmoC-phage_1 (LR_4) | 365,676 | 36 | 467 | 4 | Complete | Gamma- |
| | | Lake_Rotsee_pmoC-phage_1 (LR_5) | 341,475 | 36 | 463 | 15 | Complete | Gamma- |
| | | Lake_Rotsee_pmoC-phage_1 (LR_6) | 314,403 | 38 | 442 | 0 | Complete | Gamma-[c] |

[a]The taxonomy is determined based on *pmoC* phylogeny including both phage and bacterial *pmoC* genes. [b]Highly similar phage genomes, with identical large terminase and DNA polymerase sharing >99.5% amino-acid similarity. [c]Fragmented *pmoC*. [d]Some cells within the population only have partial *pmoC*. [e]Partial *pmoC*.

analyses (see Methods). Phylogenetic analyses based on the concatenated sequences of 13 universal phage-specific proteins determined by protein family analyses (Fig. 3 and see also Supplementary Table 6) and DNA polymerases (see Supplementary Fig. 19) suggested that all pmoC-phages are *Myoviridae*. Generally, the more similar the phage genome size the closer their phylogenetic relationship.

**Predicted hosts of pmoC-phages.** CRISPR (clustered regularly interspaced short palindromic repeats)–Cas analyses found that one spacer of Methyloparacoccus_57, and another spacer of a published *Methylobacter* genome, targeted the pmoC-phage BML_4 (see Supplementary Fig. 20). However, none of the other pmoC-phage genomes was targeted by a spacer from any CRISPR system identified. Thus, we predicted their hosts using the similarity between the sequences of PmoC in phages and coexisting bacteria, assuming that the phage-associated *pmoC* genes were acquired by lateral transfer from their bacterial hosts[7,33,34] (Fig. 2b). Methyloparacoccus_57 was predicted as the host for the four Canada oil sands pmoC-phages (Table 1). The co-occurrence of Methyloparacoccus_57 and PDSYNTPWS_1 (without *pmoC*), which is highly similar to BML_3, in the heavy PDSYNTPWS DNA-SIP fraction supports this. In LM, CB and LR samples, alphaproteobacterial and gammaproteobacterial

methanotrophs were predicted as hosts of the pmoC-phages. One predicted host, *Methylocystis* sp. (an alphaproteobacterium), and the infecting pmoC-phages LM_1, LM_2, LM_3, LM_5 and LM_8, were detected together in all 5 years, especially in samples collected in September/October of each year (see Supplementary Fig. 21). The phages LM_4 and LM_6 and their predicted gammaproteobacterial hosts (*Methylobacter* sp. and *Methyloparacoccus* sp., respectively) coexisted in the communities collected in 2012. The pmoC-phages from Crystal Bog were predicted to replicate in *Methylocystis* sp. (CB_1, CB_2, CB_4 and CB_5) and *Methyloparacoccus* sp. (CB_3), and time-series analyses verified that they coexisted in the communities (see Supplementary Fig. 21). The Lake Rotsee pmoC-phages were predicted to infect *Methylocystis* sp. (LR_1 and LR_2), *Methyloparacoccus* sp. (LR_3) and *Methylobacter* sp. (LR_4, LR_5 and LR_6). Together, these results strongly support the predicted host–phage relationships.

**Metabolic potentials of pmoC-phages and their relatives.** We evaluated the protein families of pmoC-phages and related phage genomes to determine whether PmoC is associated with any other specific protein(s) (Fig. 4 and see also Supplementary Table 7). We found that PmoC is the only protein specific to all pmoC-phages

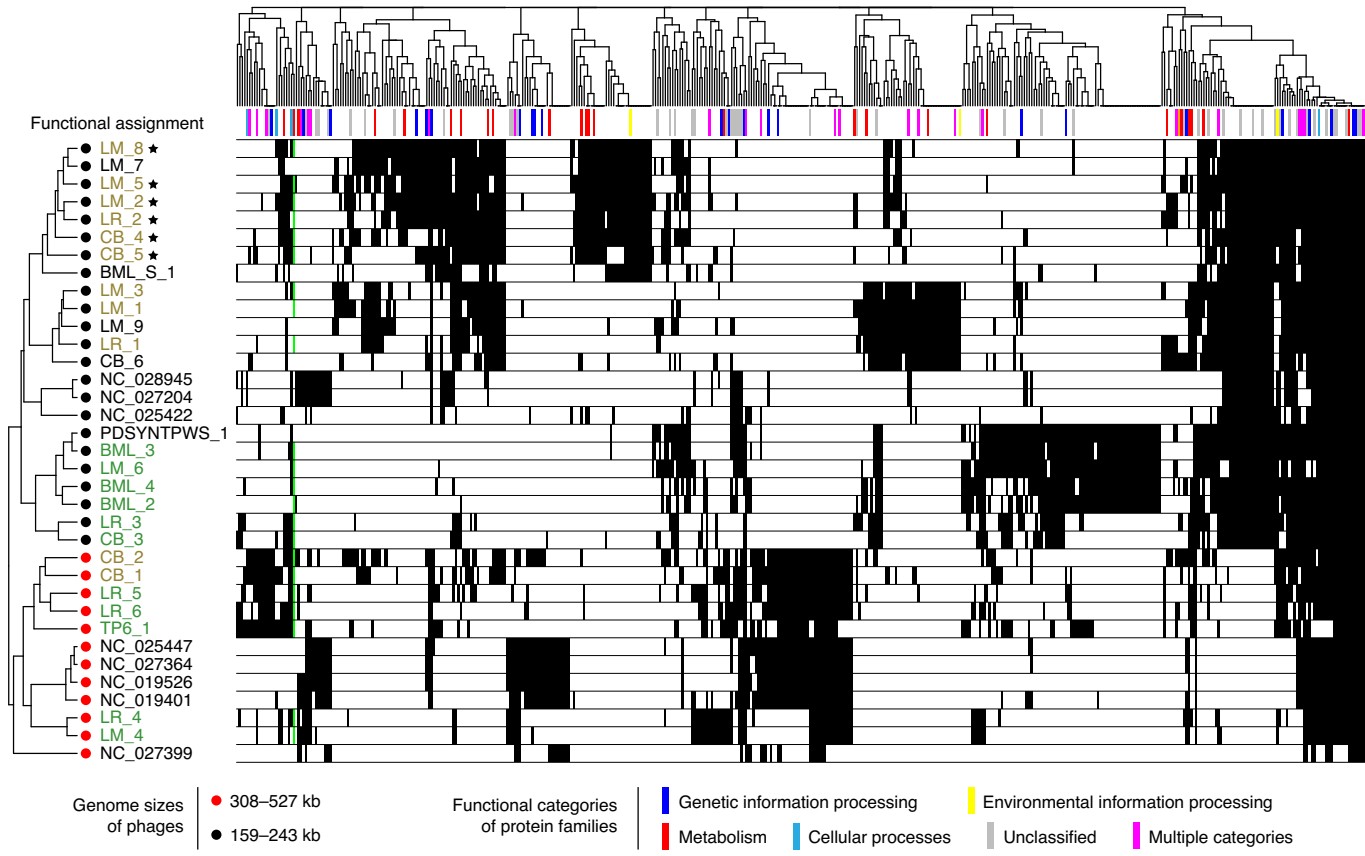

Functional assignment

**Fig. 4 | Metabolism of pmoC-phages and their relatives.** Clustering of phages based on the presence/absence profiles of protein families that are encoded by at least five phages. The phage-associated PmoC is highlighted by a green bar. The names of pmoC-phages infecting alpha- and gammaproteobacterial methanotrophs are shown in grey and green, respectively. The pmoC-phages with *pmoC* and HSP20 genes next to each other are indicated by stars (see Supplementary Table 7 for details).

(Fig. 4). Genes for heat shock protein HSP20 were detected in all but three pmoC-phages and are encoded next to *pmoC* in six pmoC-phages. However, the significance of this is difficult to evaluate because HSP20 has been reported as a core gene of cyanobacteria phages (cyanophages)[35], which are phylogenetically related to the phages reported here (see Supplementary Fig. 19). Moreover, all five related phages without *pmoC* also encode HSP20, suggesting that HSP20 may not be related to the PmoC function (Fig. 4). HSP20 is a small heat shock protein that may improve the survival of the host bacteria when they are challenged by elevated temperature, although it also has been suggested that HSP20 might be important for scaffolding during maturation of the capsid[35].

It is of interest that *cofF*, required for the biosynthesis of coenzyme F420 involved in methane metabolism, is encoded by CB_1 and CB_2 (both are pmoC-phages) (see Supplementary Fig. 22). Other genes relevant to host metabolism were detected in subsets of phages (see Supplementary Figs. 22 and 23 and also Supplementary Information).

**Transcriptional analyses of pmoC-phages.** Of the six pmoC-phages from Lake Rotsee, LR_4, LR_5 and LR_6 (genome sizes >300 kb; see Table 1) showed high transcriptional levels indicative of replication at the time of sampling in November and December 2017. Transcript data indicate that only LR_4 was highly active in the January 2018 sample (Fig. 5a and see also Supplementary Fig. 24). The three pmoC-phages with smaller genomes (159–168 kb; see Table 1) were probably inactive, based on the mapping of only a few RNA reads to their genomes. It is interesting that the *pmoC* genes of LR_4, LR_5

and LR_6 were highly expressed (generally among the top 20 most active genes), as were genes encoding phage DNA packaging and particle assembly-related proteins, including major capsid, prohead, phage tail, tail fibre, tail sheath and scaffolding proteins (Fig. 5b–d). Given that structural genes are generally expressed late in replication, we interpret the co-expression pattern to indicate that *pmoC* is important during the late phase of phage replication. It should be noted that the *pmoC* of LR_6 is predicted to be fragmented, and the C terminus was much less expressed compared with the N terminus (which contains the active site). The non-coding region of unknown function between the *pmoC* gene fragments was transcribed at a low level (see Supplementary Fig. 25). The bacterial host of LR_4, LR_5 and LR_6, a *Methylobacter* sp., showed much higher growth rates (determined by iRep values) than the hosts of the inactive pmoC-phages and bacterial methanotrophs in the same community that were not infected by pmoC-phages (Fig. 5e). In summary, these results indicate the potential significance of the phage-associated *pmoC* genes for pmoC-phages during infection, and support our inference that phage-associated *pmoC* can impact overall rates of methane oxidation in freshwater ecosystems.

## Discussion
**PmoC-phages were overlooked in previous analyses.** Previous cultivation-based studies isolated phages of bacterial methanotrophs from various habitats including oil waters and soil, but genomes of these phages have not been reported[36,37]. To date, only 13 genome scaffolds of phages infecting methanotrophs (10–62 kb in length) have been reported, and these sequences came from

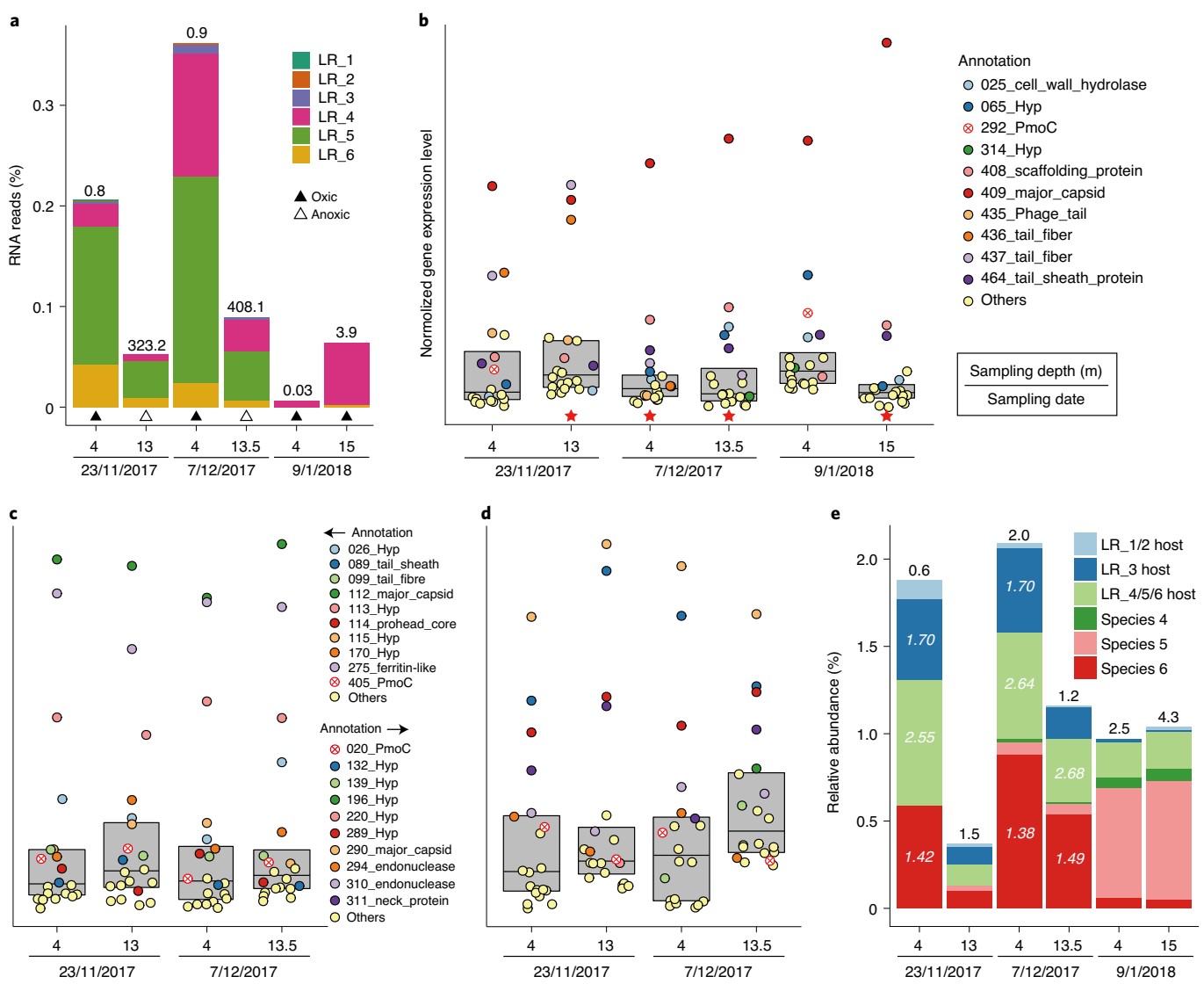

**Fig. 5 | Transcriptional analyses of pmoC-phages and information about bacterial methanotrophs in Lake Rotsee. a**, The percentage of RNA reads mapped to the pmoC-phages. The concentration of methane (in µM) is shown above the bar. **b–d**, The 20 most highly expressed genes of LR_4 (**b**), LR_5 (**c**) and LR_6 (**d**). Only the functional predictions for the top 10 genes are listed. When *pmoC* genes are within the top 20 most highly expressed genes, they are indicated by circles containing a red x. Red stars indicate that the *pmoC* gene was expressed, but not one of the 20 most highly expressed genes. Box plots enclose the first to third quartiles of data values, with a black line at the median value. **e**, The relative abundances of methanotrophs in each of the six samples. The total cell count ($\times 10^5$ cells ml$^{-1}$) of the methanotrophs in each sample is shown above the bar. The iRep values indicating growth rates are shown for the methanotrophs when a given genome has ≥5× coverage in the corresponding sample. Hyp, hypothetical protein.

thawed permafrost samples[38]. In the present study, we described 22 large genomes of pmoC-phages (up to 527 kb; Table 1) which we propose can infect bacterial methanotrophs, but none of them is genomically or phylogenetically related to those from permafrost. The pmoC-phages have been overlooked in previous studies, in part because of the focus on high-level patterns such as global distribution, diversity and host specificity rather than gene inventories[39], and in part because of the high similarity between phage-associated and bacterial PmoC fragment assemblies. The reconstruction of pmoC-phage genomes from multiple distinct habitats highlights the power of genome-resolved metagenomics and also the necessity of manual genome curation for accuracy[40].

**Why only *pmoC* in phages?.** As analysed in the present study, *pmoC*, but not *pmoA* and *pmoB*, subunits of pMMOs were detected in phages. Similarly, previous studies reported that *amoC* was the

only subunit of ammonia monooxygenase (homologue of pMMO) encoded by phages infecting Thaumarchaeota[6,7]. Although possibly acquired from bacterial hosts along with other genes, only *pmoC* was retained because it can enhance phage fitness alone. For substrate binding and oxidation of methane in bacterial methanotrophs, there is increasing evidence indicating the essential role of PmoC, but the absolute necessity of PmoB is questionable (nevertheless it is important)[15,29,30,41]. Given that the structure of PmoC is largely disordered when the cell membrane is perturbed[42], we suggest that the additional *pmoC* genes, encoded in either the bacterial methanotroph or phage genomes (see Supplementary Table 3), could augment methane oxidation. Although we do not have data to constrain how this occurs, it seems reasonable to speculate that it may sustain methane oxidation under abnormal environmental conditions. The availability of an alternative enzyme may also be beneficial when metals used in the normal bacterial subunit are in

low abundance, given that Zn or Cu can be used in the PmoC catalytic site. Regardless of how the phage-associated *pmoC* functions in detail, the promotion of methane oxidation is probably beneficial to the phage via the provision of NAD[+] needed for replication[43].

Previous studies suggested that different copies of *pmoC*[15,44,45] and *amoC*[46] from a single organism have distinct expression preferences under different conditions. Condition-dependent expression of *pmoC* is probably determined by sequence divergences in their termini, given that the middle regions are generally very similar (see Supplementary Fig. 5). As the bacterial and phage-associated PmoC sequences are also generally divergent in the N and C termini (Fig. 2a, and see also Supplementary Fig. 10), the phage-associated PmoC may function as the stand-alone PmoC in bacterial methanotrophs under some conditions. Relatedly, sequence divergences in termini of phage-associated PmoC may increase the fitness of pmoC-phages after infection. Our findings motivate the biochemical investigation of the role of phage-derived PmoC in the functioning of pMMOs.

**Potential biogeochemical impacts of pmoC-phages.** Generally, the predicted hosts were eliminated after the appearance of the infecting pmoC-phages (for example, LM samples; Supplementary Fig. 21), suggesting that the pmoC-phages could reduce methane oxidation in an ecosystem by lysing their bacterial methanotroph hosts. On the other hand, pmoC-phages may accelerate methane oxidation, as noted in the 'Why only *pmoC* in phages?' section. Modulation of methane oxidation rates may be important given that freshwater lake ecosystems are important sources of terrestrial methane emission[17,47].

The presence of photosynthesis genes in pmoC-phage genomes is intriguing (see Supplementary Fig. 23), given that they probably had to infect a cyanobacterial cell to acquire them. The pmoC-phages are phylogenetically related to cyanophages (see Supplementary Fig. 19), so they may replicate in cyanobacteria under conditions when they co-occur with them. The large genome size compared with most phages known to date may include genes required for host range expansion. Given that cyanobacteria produce $O_2$ that is required for methane oxidation by methanotrophs, and that a very recent study indicated the production of methane by cyanobacteria[48], it is possible that future work will show that pmoC-phages with a broad host range can have far-reaching impacts on the methane cycle.

## Conclusion

Our analyses suggest that some phages with large genomes that infect methanotrophs have *pmoC* (pmoC-phages), and so have the potential to impact methane oxidation as well as the carbon cycle. The phage-associated *pmoC* appears to be most active during late infection and the infected bacteria exhibit the fastest growth rates of methanotrophs in the system, supporting the inference that pmoC-phages can increase methane oxidation rates in freshwater ecosystems.

## Methods

**Sampling, DNA extraction and metagenomic analyses.** The BML samples were collected from multiple depths of an end pit lake for oil sands waste remediation in Alberta, Canada from 2015 to 2018 (see Supplementary Table 1). The geochemical features of the samples were determined in situ or in the laboratory as previously described[21]. Genomic DNA was collected filtering approximately 1.5 l water through 0.22-μm Rapid-Flow sterile disposable filters (Thermo Fisher Scientific) and stored at −20 °C until DNA extraction. DNA was extracted from the filters as previously described[49]. The DNA samples were purified for library construction and sequenced on an Illumina HiSeq1500 platform with paired-end 150-bp kits. The LM and CB samples were collected from Lake Mendota from 2008 to 2012 and Crystal Bog from 2007 to 2009 (see Supplementary Table 8). The geochemical features and the procedures of sampling, DNA extraction and sequencing were detailed elsewhere[50], and the metagenomic reads were reassembled for pmoC-phages and their host in the present study, as well as six metagenomic and their corresponding metatranscriptomic datasets from Lake Rotsee (47° 04′ 11″ N,

8° 18′ 51″ E)[16]. The raw reads of each metagenomic or metatranscriptomic sample were filtered to remove Illumina adaptors, PhiX and other contaminants with BBTools[51], and low-quality bases and reads using Sickle (v.1.33; https://github.com/najoshi/sickle). The high-quality reads of each metagenomic sample were assembled using idba_ud[52] (parameters: --mink 20 --maxk 140 --step 20 --pre_correction). For a given sample, the high-quality reads of all samples from the same sampling site were individually mapped to the assembled scaffold set of each sample using Bowtie2 with default parameters[53]. The coverage of a given scaffold was calculated as the total number of bases mapped to it divided by its length. Multiple coverage values were obtained for each scaffold to reflect the representation of that scaffold in the related samples collected from the same site. For each sample, scaffolds with a minimum length of 3 kb were assigned to preliminary draft genome bins using MetaBAT with default parameters[54], with both tetranucleotide frequencies and coverage profiles of scaffolds considered. The scaffolds from the obtained bins and the unbinned scaffolds with a minimum length of 1 kb were uploaded to the ggKbase platform. The protein-coding genes were predicted using Prodigal[55] (-m -p meta) from scaffolds and annotated using usearch[56] against KEGG[57], UniRef[58] and UniProt[59]. The genome bins determined by MetaBAT were manually modified at ggKbase based on the consistency of GC content, coverage and taxonomic information of the scaffolds, and the scaffolds identified as contaminants were removed. The modified genome bins were validated based on the coverage profiles of the scaffolds. The tRNA genes were predicted using tRNAscanSE[60] and 16S rRNA genes with HMM databases as previously described[61].

**Reanalysis of published oil sands datasets.** Datasets from four published studies of oil sands waste lakes were reanalysed in the present study.

*Study 1.* First, we analysed the datasets from enrichments amended with a short-chain alkane ($C_6$–$C_{10}$), naphtha or toluene[27]. We did not detect Methyloparacoccus_57 or any pmoC-phage in these enrichments. Second, we analysed the other two metagenomic datasets used for comparison in the original paper, that is, TP6 and TP_MLSB. The sample TP6 (UTM 466358E 6319838N) was collected in 2012 from Suncor tailings pond at the depth of 6 m and sequenced with both 454 pyrosequencing and Illumina (National Center for Biotechnology Information (NCBI) accession no. SRX327722). We detected one pmoC-phage genome (referred to as 'TP6_1') from the original assembly and extended it using the 454 pyrosequencing and Illumina reads to generate the current version (see Table 1). None of the pmoC-phages identified in Syncrude BML samples was detected in this sample. For its host, we compared the PmoC sequence of TP6_1 with all other PmoC sequences from the assembly, and analysed all the bacterial and archaeal species in the community via ribosomal protein S3 (rpS3) phylogeny for methanotrophs, and found that the host of TP6_1 is Methyloparacoccus_57. The sample TP_MLSB was collected from Syncrude in 2011 (NCBI accession no. SRR636569), and the quality Illumina reads were downloaded and mapped to genomes reconstructed from Syncrude BML (with >98% nucleotide identity). As a result, Methyloparacoccus_57 (sequencing coverage: 7.37×; genome covered: 97.8%), pmoC-phages of TP6_1 (sequencing coverage: 5.24×; genome covered: 97.8%; see Extended Data Fig. 2a) and BML_3 (sequencing coverage: 7.01×; genome covered: 89.6%) were detected (see Extended Data Fig. 2b). We did not assemble this dataset to recover the genomes because of the low sequencing coverage. It is interesting that the *pmoC* region of BML_3 was mapped by only two reads, indicating that the corresponding phage in TP_MLSB generally did not contain *pmoC*. The read pile-ups (abnormally high coverage) may indicate the existence of other related phage(s) and/or repeat regions.

*Study 2.* Saidi-Mehrabad et al.[26] collected surface water (0–10 cm) at 1- to 3-month intervals over 2010–2011 from two tailings ponds near Fort McMurray, Alberta, Canada (that is, Pond A and Pond B as designated in the original paper). As described in the original paper, 'An aerobic methanotroph belonging to the *Methylococcus/Methylocaldum* cluster of Gammaproteobacteria (OTU12103) was among the predominantly detected OTUs in Pond A, making up on average 1.5% of all reads', and so de novo assembly of the metagenomic dataset was performed of PD_SYN_TP_WS_002_003_071511 (NCBI accession no. SRX327520; referred to as 'PDSYNTPWS' hereafter) sequenced by Illumina, and found that the predominant OTU12103 corresponds with Methyloparacoccus_57 reported in the present study. In fact, the 16S rRNA gene sequence from their assembly was identical to that of Methyloparacoccus_57. Phylogenetic (based on rpS3) and sequencing coverage analyses also indicated that Methyloparacoccus_57 is the most abundant bacterial methanotroph in the community (see Extended Data Fig. 1a). Binning and subsequent curation yielded the Methyloparacoccus_57-related genome from PDSYNTPWS, referred to as 'Methyloparacoccus_57_PDSYNTPWS'. The Illumina reads of PDSYNTPWS were mapped to the pmoC-phage genomes of BML_2, BML_3, BML_4 and TP6_1 (see 'Study 1' section). This revealed the presence of phages similar to BML_3 (see Extended Data Fig. 2c). Manual curation of the corresponding scaffolds generated a high-quality genome (referred to as PDSYNTPWS_1). The genomic alignments of PDSYNTPWS_1 and BML_3 (and LM_6 as well) are shown in Supplementary Fig. 16 and described in Results. In addition, DNA-SIP analyses with ¹³CH$_4$ were conducted to track the active methane

oxidizers in the PDSYNTPWS sample[26]. A 'Five microliters of a selected "heavy" SIP fraction' of DNA sample was collected for amplification and sequencing for metagenomic analyses. The resulting Illumina reads (382 million read pairs) were downloaded and mapped to the genomes of Methyloparacoccus_57_PDSYNTPWS and PDSYNTPWS_1. As a result, ~6.58% of the reads could be mapped to Methyloparacoccus_57_PDSYNTPWS (see Extended Data Fig. 3a), and a small fraction of reads was mapped to PDSYNTPWS_1 (see Extended Data Fig. 3b). The uneven depth across the scaffolds may be due to the multiple displacement amplification used in DNA preparation. We also performed de novo assembly of the DNA-SIP data and obtained a total length of 90-Mbp scaffolds. Phylogenetic analyses based on rpS3 indicated that Methyloparacoccus_57_PDSYNTPWS and some other gammaproteobacterial methanotrophs in the community were actively oxidizing methane (see Extended Data Fig. 3c). Saidi-Mehrabad et al.[26] also reported a total of 22 16S rRNA gene datasets (sequenced by 454 GS FLX Titanium) in the original paper, 16 of which could be downloaded from NCBI Sequence Read Archive (SRA) via the accession no. provided (SRP013946). The 16S rRNA gene sequences were searched against that of Methyloparacoccus_57_ PDSYNTPWS using BLASTn (>98% similarity, >500 alignment length), and the total number of hits and the relative abundances were calculated for each sample. As we could not match the NCBI SRA datasets to the samples described in the original paper, we show the SRA accession no. and sample description as well (see Extended Data Fig. 1b).

*Study 3.* A total of 12 metagenomic datasets (sequenced by 454 pyrosequencing or Illumina) from oil sands-related habitats were reported by An et al.[25]. Methyloparacoccus_57 was detected in only PDSYNTPWS (454 pyrosequencing reads) by a 16S rRNA gene sequence search. Also, genomic fragments similar to phage PDSYNTPWS_1 were identified in the sample (see Extended Data Fig. 2d). However, these fragments covered only a small part of the genome, suggesting a low abundance of the corresponding phage in the sample.

*Study 4.* Oil sands process-affected water (OSPW) was collected in 2012 for incubation experiments that involved the addition of benzene or naphthalene, to reveal the microorganisms in OSPW responsible for compound degradation[28]. One control water sample was also analysed via 16S rRNA gene sequence analyses (sequenced by 454 GS FLX Titanium). The 16S rRNA gene sequence datasets were downloaded from NCBI SRA via the accession no. SRP109130 provided in the original paper, and compared against that of Methyloparacoccus_57 reported in the present study by BLASTn (>98% similarity, >500 alignment length). The analyses indicate that Methyloparacoccus_57 was not the primary consumer of naphthalene or benzene; however, it was highly abundant in the natural and treatment control OSPW samples (see Extended Data Fig. 1c), indicating the prevalence of these bacteria in situ.

**Relative abundance and growth rate analyses.** The rpS3 was used as a taxonomic marker gene for microbial community composition analyses. All the rpS3 proteins were predicted using hmmsearch[62] based on the tigrfam[63] HMM databases (TIGR01008 for Archaea and Eukaryotes, and TIGR01009 for Bacteria). The HMM hits were filtered by the tigrfam cutoff and searched against the NCBI RefSeq database[64] by BLASTp to remove those with the best hit of Eukaryotes. The retained bacterial and archaeal rpS3 amino-acid sequences were clustered by cd-hit[65] with 100% similarity (-c 1, -aL 0.8, -aS 0.8, -G 0). The nucleotide sequences of all representative rpS3 proteins were extracted and used as a dataset for reads mapping to calculate their coverage in each sample, which was performed by Bowtie2 (ref. [66]) with the default parameters. The coverage of a given scaffold was reported only when the reads from a given sample covered at least 50% of the nucleotide sequence. The relative abundance of a taxon in a given sample was calculated as the coverage of the corresponding rpS3 divided by the collective coverage of all representative rpS3 proteins in the sample. The growth rate of a given species was determined using iRep[67] based on the read mapped to the corresponding curated genome (≥5× coverage) with a maximum of one mismatch per read.

**Manual genome curation of genomes.** The phage scaffolds were identified using ggKbase based on the presence of phage-specific genes as previously described[68], including capsid, phage, virus, prophage, terminase, prohead, tape measure, tail, head, portal, DNA packaging, the presence of genes similar to previously identified phage-associated genes of unknown function and lack of many host-specific genes. The protein-coding genes of phage scaffolds were searched against the HMM databases of proteins involved in methane metabolisms. The phage scaffolds with *pmoC* genes and also a minimum sequencing coverage of 20× were manually curated to completion. This involved circularization, filling of scaffolding gaps and fixing of any local assembly errors[40]. Manual correction of local assembly errors and extension of phage scaffolds were time-consuming but essential to reveal their metabolic potentials and confirm the absence of other pMMO subunits. In detail, first, a given phage scaffold with *pmoC* was extended using unplaced paired reads in Geneious[69]. Local assembly errors that were identified based on lack of perfect support by mapped reads were manually fixed. Second, the extended fragments were searched against the whole assembled scaffold set for the potential missing parts of the phage genome, the retrieved scaffolds were assembled with

the extended phage scaffold and, then, the overall assembly was confirmed by read mapping. Scaffold extension and addition of missing scaffolds were continued until a circular phage genome was obtained. All the curated phage genomes were verified by mapping the reads to the final genomes. Exceptionally, the scaffold of pmoC-phage TP6_1 was sequenced by 454 pyrosequencing and Illumina[25] and extended by overlap at the ends of scaffolds detected by BLASTn using 454 reads, followed by confirmation of the extension by Illumina reads. The BLASTn search and extension were performed several times until no more scaffolds with end overlap could be found. For the genomes of phages closely related to pmoC-phages, we first identified the scaffolds by searching against all the large terminase proteins from already reconstructed pmoC-phages, and those scaffolds having a large terminase with ≥80% amino-acid similarity were selected as targets for manual scaffold extension and curation to completion. The similarity of phage genomes was calculated using the online average nucleotide identity tool[70]. For genomes of bacterial methanotrophs, all the local assembly errors except those detected in pMMO-encoding regions (see 'Manual genome curation of genomes' section) were checked and fixed by ra2.py[61].

**Confirmation of Methyloparacoccus_57 in all BML samples.** When the biomass of a given population accounts for only a small fraction of that of a collected sample, de novo metagenomic assembly and subsequent analyses may not be able to detect the population. In the present study, the host–phage relationship was predicted based on the similarity of the PmoC sequences (among those from pmoC-phages and bacterial methanotrophs), followed by evaluation of the co-occurrence of phage and its predicted host (based on their genomic sequences assembled from metagenomic data). The pmoC-phages of BML_2, BML_3 and BML_3 were predicted to replicate in Methyloparacoccus_57; however, assembled fragments of this population could be detected in only 14 of the 28 analysed BML samples. To test for the existence of Methyloparacoccus_57 in the other 14 BML samples (from which the rpS3 of Methyloparacoccus_57 was not assembled), we first curated the genome of Methyloparacoccus_57 (2,444,800 bp in length) from the sample of BML_10242017_9.75m (which has the highest sequencing coverage of this population), then the quality reads from all BML samples were individually mapped to the curated Methyloparacoccus_57 genome, with two mismatches allowed for each mapped read (that is, >98.6% nucleotide similarity). As expected, for the samples with Methyloparacoccus_57 fragments assembled, the number of reads (18,252–556,226 reads) mapped to a scaffold strongly correlates with the length of the corresponding scaffold (see Supplementary Fig. 7a, sample names in black). For the 14 BML samples without Methyloparacoccus_57 rpS3 assembled (see Supplementary Fig. 7a, sample names in red), only 380–6,046 reads were mapped to the curated Methyloparacoccus_57 genome (and the number of reads mapped to a scaffold also strongly correlated with the length of the corresponding scaffold). We also mapped reads to the scaffolds (that is, BML_10242017_9_75m_ scaffold_435) with the ribosomal proteins (see Supplementary Fig. 7b), and found all samples had reads mapped to this region. In summary, we concluded that Methyloparacoccus_57 was in all the 28 analysed BML samples, although some of them at very low abundance.

**Bacterial sMMO and pMMO subunits.** To reveal the sMMO and pMMO subunits in the published genomes of bacterial methanotrophs, all the genomes assigned to the well-known methanotroph genera[71] were downloaded from NCBI (see Supplementary Table 3), along with their protein sequences and annotation information. The stand-alone *pmoC* genomes in published genomes were identified manually, based on their genomic context. Those located at the end of scaffolds were assigned as 'questionable stand-alone'. For the bacterial methanotrophs with genomes reconstructed in the present study, their sMMO and pMMO subunit genes were identified based on functional predictions (see 'Sampling, DNA extraction and metagenomic analyses' section). The corresponding scaffolds were checked for potential assembly errors by read mapping and careful manual curation was performed if an error was identified. Local assembly errors occurred primarily due to the high sequence similarity of bacterial (multiple copies of pMMO operons and also stand-alone *pmoC*) and phage-associated *pmoC* genes. For the pMMO operons and stand-alone *pmoC* scaffolds of the bacterial methanotrophs, we generally manually curated them using the reads from the samples without pmoC-phages detected.

**CRISPR–Cas analyses.** All the predicted proteins of scaffolds with a minimum length of 1 kb were searched against local HMM databases, including all reported Cas proteins, and the nucleotide sequences of the same set of scaffolds were scanned for CRISPR loci using minced[72] (-minSL = 17). The spacers were extracted from the scaffolds with CRISPR loci as determined by minced, and also from reads mapped to these corresponding scaffolds using the python script (crispy.py) as previously described[68]. For the published methanotroph genomes (see above), only spacers from the scaffold consensus sequences were extracted, because no mapped reads are available. Duplicated spacers were removed using cd-hit-est (-c 1, -aS 1, -aL 1) and the unique spacer sequences were used to build a database for BLASTn searches (task = blastn-short, e-value = $1 \times 10^{-3}$) against the pmoC-phage genomic sequences. Once a spacer was found to target a pmoC-phage scaffold (≥30 bp), the original scaffold of the spacer was checked for a CRISPR locus and Cas proteins.

**Distribution of phages and their predicted hosts.** The quality reads from each sampling site were mapped to the genomes of pmoC-phages reconstructed from the same site. The occurrence of a given phage in a given sample was determined if ≥75% of its genome could be covered by reads with ≥95% nucleotide similarity. The sequencing coverage of a given pmoC-phage in a sample was calculated using the total length of mapped reads divided by the length of the phage genome. The occurrence of a given predicted host was established if all the scaffolds were mapped by reads with ≥98% nucleotide similarity and ≥75% of the scaffold was covered. The sequencing coverage of a given scaffold in the host genome was determined as for pmoC-phage genomes, and the average sequencing coverage of all the scaffolds in the genome was calculated and used as the host genome coverage. If a high-quality genome bin for a predicted host could not be reconstructed, the host coverage was determined as that of the scaffold with the pMMO operon (see above for the determination of the pmoC-phages and their predicted host in published oil sands-related metagenomic datasets). Methyloparacoccus_57 could be detected in LM samples (with identical 16S rRNA gene sequence found) but with very low sequencing coverage, so no quality genome was obtained. Given the high similarity between LM_6 and BML_3, we predicted that Methyloparacoccus_57 was the host of LM_6, and the genome of Methyloparacoccus_57 from BML was used to profile its presence in the LM samples, as described above.

**Phage protein family analyses.** Protein family analyses were performed as previously described[73]. In detail, first, all-versus-all searches were performed using MMseqs2 (ref. [74]), with parameters set as e-value = 0.001, sensitivity = 7.5 and cover = 0.5. Second, a sequence similarity network was built based on the pairwise similarities, then the greedy set cover algorithm from MMseqs2 was performed to define protein subclusters (that is, protein subfamilies). Third, to test for distant homology, we grouped subfamilies into protein families using an HMM–HMM comparison procedure as follows: the proteins of each subfamily with at least two protein members were aligned using the result2msa parameter of MMseqs2, and HMM profiles were built from the multiple sequence alignment using the HHpred suite[75]. The subfamilies were then compared with each other using hhblits[76] from the HHpred suite (with parameters -v 0 -p 50 -z 4 -Z 32000 -B 0 -b 0). For subfamilies with probability scores ≥95% and coverage ≥0.5, a similarity score (probability × coverage) was used as the weight of the input network in the final clustering using the Markov Cluster Algorithm[77], with 2.0 as the inflation parameter. Finally, the resulting clusters were defined as protein families. The clustering analyses of the presence and absence of protein families detected in the phage genomes were performed with Jaccard's distance and complete linkage.

**Phylogenetic analyses.** Phylogenetic analyses were performed for bacterial and phage-associated PmoC sequences identified from BML, BML_S, LM and CB samples, with NCBI bacterial methanotroph PmoC sequences (see above) included for references. The PmoC fragments from pmoC-phages were respectively concatenated as one. To reveal the phylogeny of phages with genomes reconstructed in the present study, sequences of 13 protein subfamilies retrieved from the protein family analyses (see Phage protein family analyses) were concatenated for analyses. In addition, the DNA polymerase (within the 13 proteins used for concatenation) was used as a single marker for phylogenetic analyses. All DNA polymerases of NCBI RefSeq viruses/phages were downloaded and used to retrieve references by BLASTp (using the DNA polymerase sequences reported in the present study as queries). The top 30 BLASTp hits were included as references.

For the phylogeny of bacterial methanotrophs, 16-concatenated ribosomal proteins (16RPs)[78], rpS3 and 16S rRNA gene sequences were used as markers. For protein-coding genes predicted by prodigal[55] from scaffolds with a minimum length of 1 kb, the 16RPs (including rpS3) were determined using an HMM-based search with databases built from Hug et al.[78]. For those scaffolds with eight or more of the 16RPs, the ribosomal proteins were individually aligned and filtered. Another tree based only on rpS3 was constructed using the same procedure. The references for both 16RP and rpS3 trees were selected from the Hug et al.[78] datasets using rpS3 BLASTp search with the top five hits included. The 16S rRNA genes were predicted via an HMM search as previously described[61], and any insertion with a minimum length of 10 bp was removed. The insertion-removed 16S rRNA gene datasets were aligned using a local version of SINA aligner[79] and filtered by trimAl to remove those columns with ≥90% gaps. The tree was built using IQtree[80] using the 'GTR + G4' model. References were selected based on a BLASTn search against the 16S rRNA gene datasets of Silva132 (ref. [81]), and the top five hits were included. For all the phylogenetic analyses with protein sequences, the proteins were aligned using Muscle[82] and filtered by trimAl[83] to remove those columns with ≥90% gaps, followed by tree building with IQtree[80] using the 'LG + G4' model, filtered sequences being concatenated for multiple protein-based analyses.

**SNP analyses of pmoC-phages.** As a case study, we investigated the population heterogeneity of the most commonly observed pmoC-phage in BML, BML_2, which was detected in 13 samples with ≥5× coverage. The reads from each sample were mapped to the genome, and SNPs were called using the inStrain package[84].

To discern the population dynamics of individual variants over time, we tested for variants that significantly changed in frequency between the sampling years of 2016 and 2017 (z-test; q < 0.05).

**Transcriptional analyses.** The metatranscriptomic RNA reads from each of the six LR samples were mapped to the nucleotide sequences of protein-coding genes that were predicted from pmoC-phages using Prodigal[55] (-m, -p = single), and filtered with shrinksam (https://github.com/bcthomas/shrinksam), allowing no mismatch. For a given pmoC-phage (i), to evaluate its gene expression profiles in a given sample (j), we calculated the gene expression level of a given gene (k) as $E_k = N_k / (L_k \times S_{ij})$, in which $E_k$ represents the expression level of gene k, $N_k$ is the number of reads mapped to gene k, $L_k$ is the length of gene k and $S_{ij}$ is the total number of reads from sample j mapped to all genes of pmoC-phage i.

**Reporting summary.** Further information on the research design is available in the Nature Research Reporting Summary linked to this article.

## Data availability

The genomes of pmoC-phages and their relatives reported in the present study have been deposited at NCBI under PRJNA645206, and are also available at Figshare (https://figshare.com/projects/pmoC-phages_in_freshwater_ecosystems/76623). The read archive and other accession information are provided in Supplementary Table 8. The pmoACB and Cas protein HMM datasets are available at http://tigrfams.jcvi.org/cgi-bin/Listing.cgi. The 16S rRNA gene HMM database is available at https://github.com/christophertbrown/bioscripts/tree/master/databases. Source data are provided with this paper.

## Code availability

The crispr.py script is available at https://github.com/linxingchen/CRISPR/blob/master/crispr.py.

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

## Acknowledgements

We thank M. J. Mayr for permission to use the metagenomic and metatranscriptomic datasets from Lake Rotsee[16] for analyses in the present study. We thank An et al.[25], Saidi-Mehrabad et al.[26], Tan et al.[27] and Rochman et al.[28] as the generators of publicly available oil sands-related datasets that were reanalysed in the present study. We thank R. Edwards for help in attempting to retrieve highly similar phage genomes in NCBI SRA datasets. The study was supported by the NSERC Canada and Syncrude Canada (grant no. CRDPJ 403361-10). We also thank the Chan Zuckerberg Biohub and the Innovative Genomics Institute at University of California, Berkeley for funding support. K.D.M. received funding from the US National Science Foundation Microbial Observatories program (no. MCB-0702395), the Long-Term Ecological Research Program (no. NTL-LTER DEB-1440297) and an INSPIRE award (no. DEB-1344254).

## Author contributions

L.X.C. designed the analyses. T.C.N. collected and prepared the BML and BML_S samples for sequencing. G.F.S. performed the methane analyses on BML and BML_S samples. T.C.N. and L.A.W. provided the DNA sequencing and the geochemical data of BML and BML_S samples. K.D.M. provided the metagenomic datasets of LM and CB samples. L.X.C. performed the metagenomic assembly, genome binning, genome annotation, phylogenetic analyses, HMM search and CRISPR–Cas analyses. L.X.C. and J.F.B. performed manual genome curation. R.M. and L.X.C. performed protein family analyses. A.C.C. performed the SNP analysis. L.X.C. and J.F.B. wrote the manuscript with input from A.C.C. All authors read and approved the final manuscript.

## Competing interests

The authors declare no competing interests.

## Additional information

**Extended data** is available for this paper at https://doi.org/10.1038/s41564-020-0779-9.

**Correspondence and requests for materials** should be addressed to J.F.B.

**Peer review information** Peer reviewer reports are available.

**a**

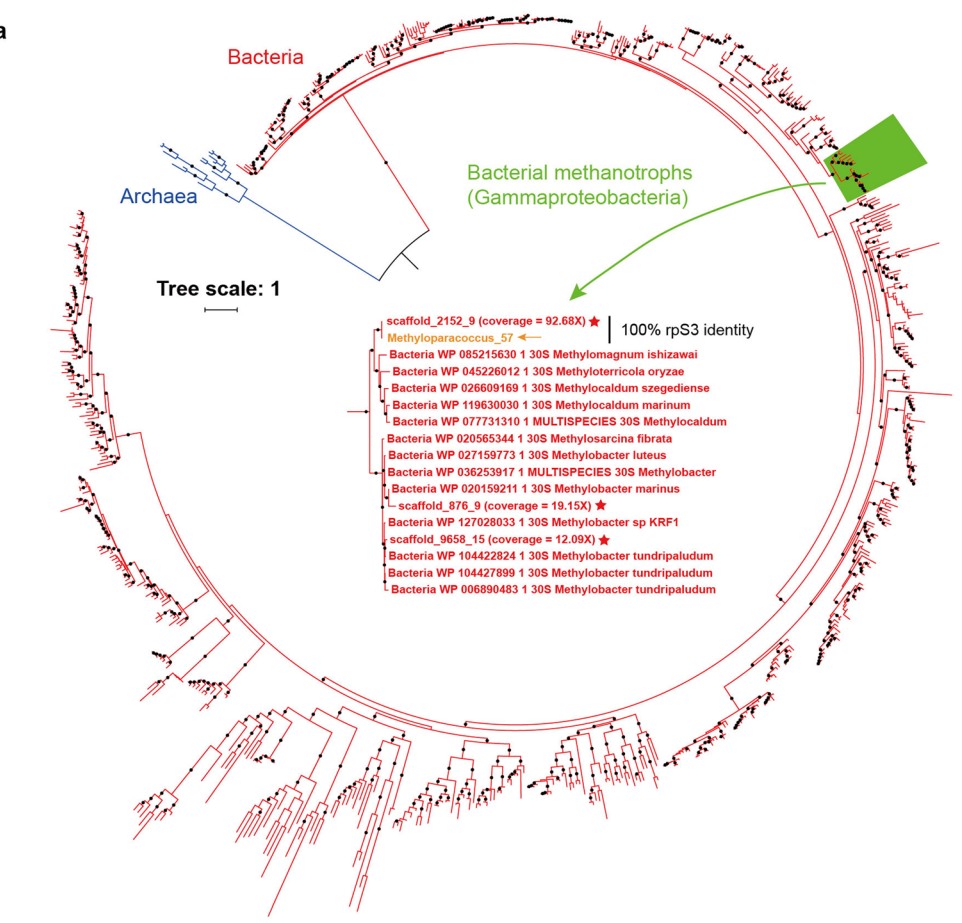

Bacteria

Archaea

Bacterial methanotrophs
(Gammaproteobacteria)

Tree scale: 1

scaffold_2152_9 (coverage = 92.68X) ★
Methyloparacoccus_57 ←
Bacteria WP 085215630 1 30S Methylomagnum ishizawai
Bacteria WP 045226012 1 30S Methyloterricola oryzae
Bacteria WP 026609169 1 30S Methylocaldum szegediense
Bacteria WP 119630030 1 30S Methylocaldum marinum
Bacteria WP 077731310 1 MULTISPECIES 30S Methylocaldum
Bacteria WP 020565344 1 30S Methylosarcina fibrata
Bacteria WP 027159773 1 30S Methylobacter luteus
Bacteria WP 036253917 1 MULTISPECIES 30S Methylobacter
Bacteria WP 020159211 1 30S Methylobacter marinus
scaffold_876_9 (coverage = 19.15X) ★
Bacteria WP 127028033 1 30S Methylobacter sp KRF1
scaffold_9658_15 (coverage = 12.09X) ★
Bacteria WP 104422824 1 30S Methylobacter tundripaludum
Bacteria WP 104427899 1 30S Methylobacter tundripaludum
Bacteria WP 006890483 1 30S Methylobacter tundripaludum

100% rpS3 identity

**b   SRA accession ID (SRA description)**        **Relative abundance (%)**

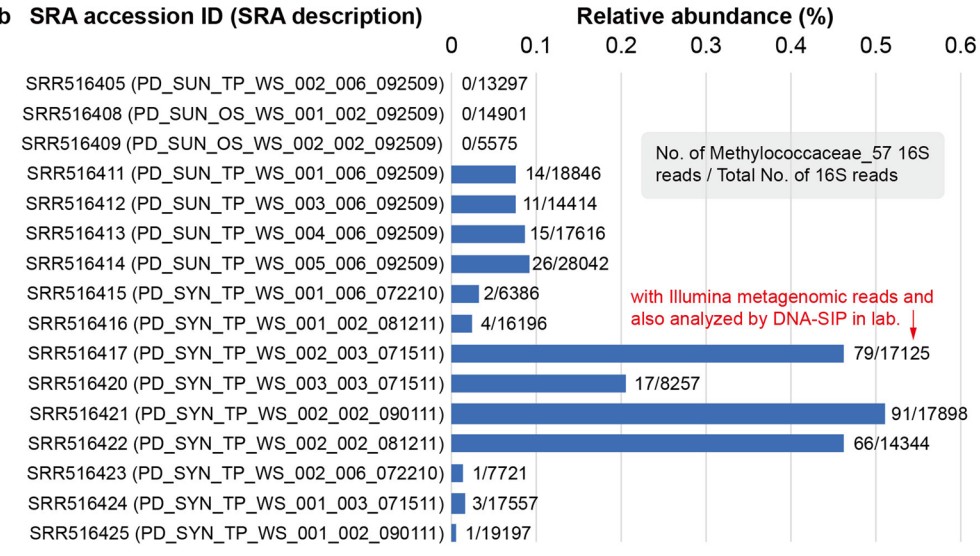

No. of Methylococcaceae_57 16S
reads / Total No. of 16S reads

with Illumina metagenomic reads and
also analyzed by DNA-SIP in lab. ↓

| SRA accession ID (SRA description) | reads |
|---|---|
| SRR516405 (PD_SUN_TP_WS_002_006_092509) | 0/13297 |
| SRR516408 (PD_SUN_OS_WS_001_002_092509) | 0/14901 |
| SRR516409 (PD_SUN_OS_WS_002_002_092509) | 0/5575 |
| SRR516411 (PD_SUN_TP_WS_001_006_092509) | 14/18846 |
| SRR516412 (PD_SUN_TP_WS_003_006_092509) | 11/14414 |
| SRR516413 (PD_SUN_TP_WS_004_006_092509) | 15/17616 |
| SRR516414 (PD_SUN_TP_WS_005_006_092509) | 26/28042 |
| SRR516415 (PD_SYN_TP_WS_001_006_072210) | 2/6386 |
| SRR516416 (PD_SYN_TP_WS_001_002_081211) | 4/16196 |
| SRR516417 (PD_SYN_TP_WS_002_003_071511) | 79/17125 |
| SRR516420 (PD_SYN_TP_WS_003_003_071511) | 17/8257 |
| SRR516421 (PD_SYN_TP_WS_002_002_090111) | 91/17898 |
| SRR516422 (PD_SYN_TP_WS_002_002_081211) | 66/14344 |
| SRR516423 (PD_SYN_TP_WS_002_006_072210) | 1/7721 |
| SRR516424 (PD_SYN_TP_WS_001_003_071511) | 3/17557 |
| SRR516425 (PD_SYN_TP_WS_001_002_090111) | 1/19197 |

**c   SRA accession ID (SRA description)**        **Relative abundance (%)**

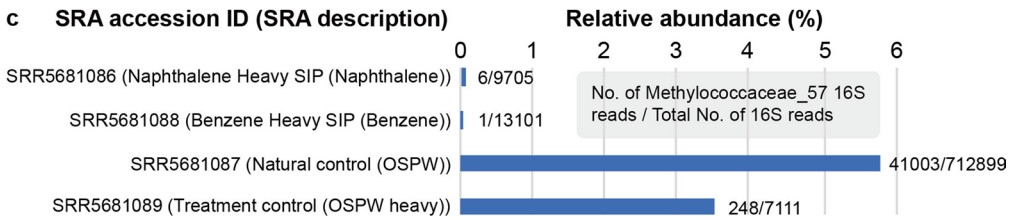

No. of Methylococcaceae_57 16S
reads / Total No. of 16S reads

| SRA accession ID (SRA description) | reads |
|---|---|
| SRR5681086 (Naphthalene Heavy SIP (Naphthalene)) | 6/9705 |
| SRR5681088 (Benzene Heavy SIP (Benzene)) | 1/13101 |
| SRR5681087 (Natural control (OSPW)) | 41003/712899 |
| SRR5681089 (Treatment control (OSPW heavy)) | 248/7111 |

**Extended Data Fig. 1 | See next page for caption.**

**Extended Data Fig. 1 | The detection of Methyloparacoccus_57 in published oil sands datasets.** (**a**) The detection of Methyloparacoccus_57 in sample PDSYNTPWS (Ref. [26]) based on ribosomal protein S3 (rpS3). The phylogeny of the methanotrophs is zoomed-in in the middle. Sequences from PDSYNTPWS are indicated by red stars. Sequencing coverages of the corresponding scaffolds are shown in the brackets, the rpS3 of Methyloparacoccus_57 (from BML) is included for reference. A black solid circles indicate bootstrap values ≥ 70. (**b**)The information of Methyloparacoccus_57 related 16S rRNA gene sequences detected in the datasets reported in Ref. [26]. (**c**) The information of Methyloparacoccus_57 related 16S rRNA gene sequences detected in the datasets reported in Ref. [28].

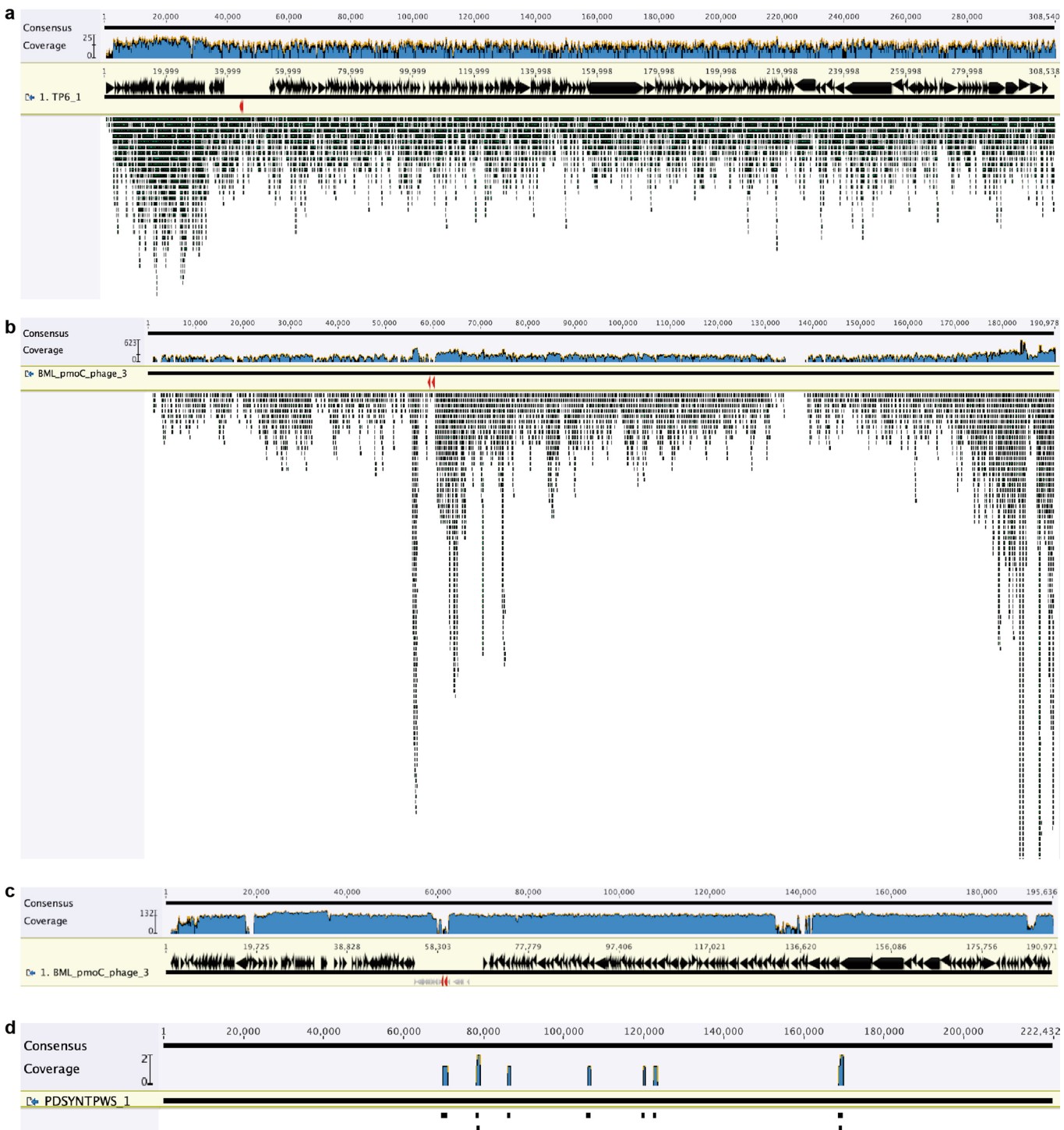

**Extended Data Fig. 2 | The detection of pmoC-phage related sequences in published oil sands datasets.** The mapping of reads from TP_MLSB (Ref. [27]) to pmoC-phage genomes of (**a**) TP6_1 and (**b**) BML_3. (**c**) The mapping of reads from PDSYNTPWS (Ref. [26]) to BML_3 indicates the existence of related phages in the sample. The *pmoC* genes are shown in red. (**d**) The alignment of 454 pyrosequencing reads from PDSYNTPWS (Ref. [25]) to phage genome of PDSYNTPWS_1. The 454 reads were reported in Ref. [25], and the phage genome of PDSYNTPWS_1 was reconstructed from Ref. [26]. The small number of reads aligned was likely due to the low sequencing coverage of 454 pyrosequencing, and/or the low abundance of related phage in the sample, or genetic divergences. The mapping was performed by Bowtie2 (Ref. [53]) and filtered allowing ≤ 2 mismatches per read (that is, ≥ 98% nucleotide sequence similarity).

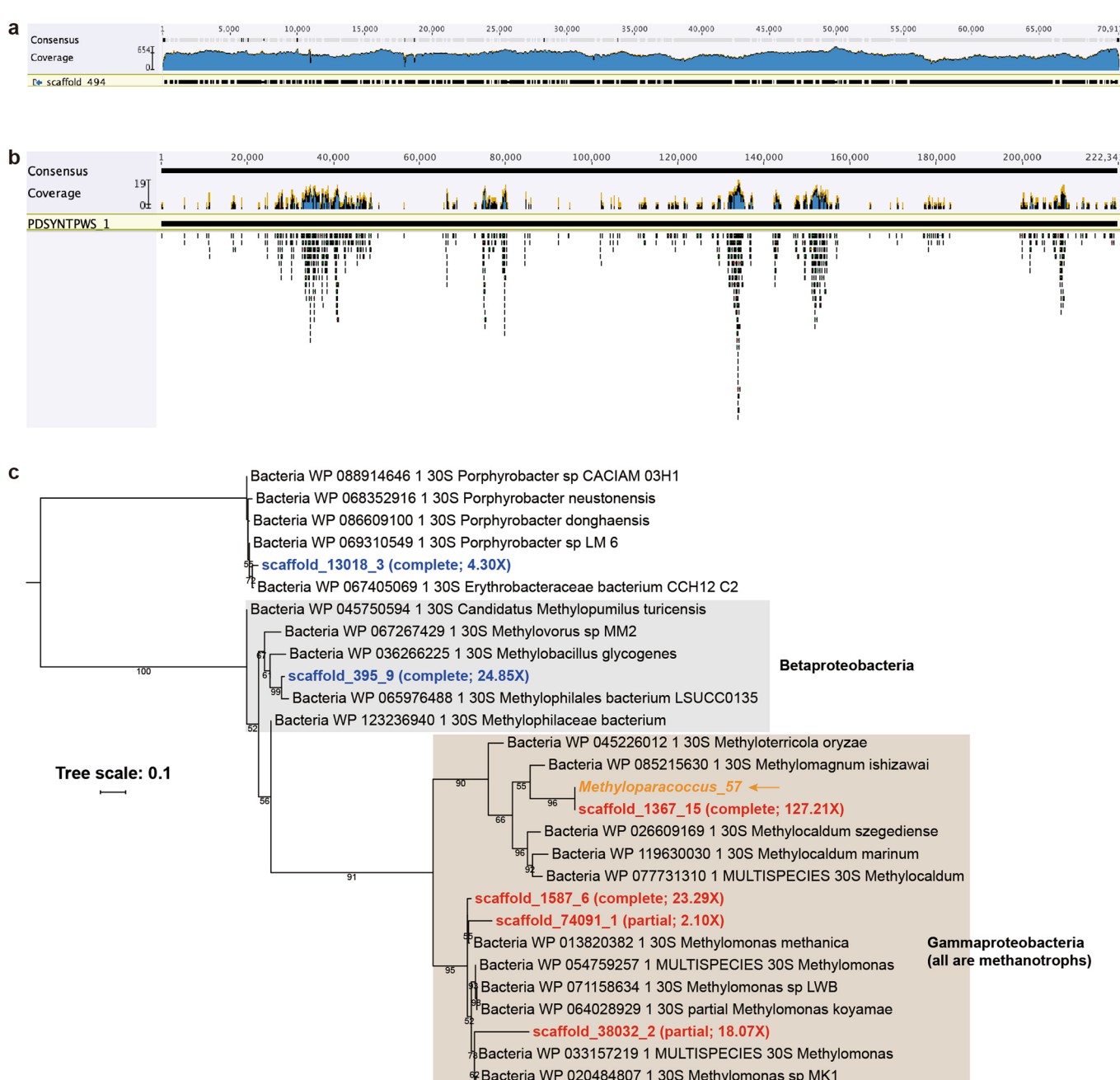

**Extended Data Fig. 3 | The reanalysis of published DNA-SIP metagenomic dataset from oil sands sample in Canada.** (**a**) Reads from the heavy DNA-SIP fraction of PDSYNTPWS mapped to the longest scaffold of Methyloparacoccus_57 (scaffold_494; 70,351 bp). The mapping was performed by Bowtie 2 and filtered to allow ≥ 98 nucleotide identity. (**b**) Reads from the heavy DNA-SIP fraction of PDSYNTPWS were mapped to PDSYNTPWS_1. The mapping was filtered to allow ≥ 98 nucleotide identity. The uneven depth may be due to the multiple displacement amplification used in sequencing sample preparation (see Ref. [26] for details). (**c**) Phylogenetic analyses showing the active members in the community that revealed by DNA-SIP analyses. The phylogeny was performed based on the rpS3 protein sequences, rpS3 of Methyloparacoccus_57 (indicated by an arrow) was included for reference. The sequencing coverages of the corresponding scaffolds are shown in the brackets. The methanotrophs are indicated in red, and non-methanotrophs in blue.

# nature research

# Reporting Summary

Nature Research wishes to improve the reproducibility of the work that we publish. This form provides structure for consistency and transparency in reporting. For further information on Nature Research policies, see Authors & Referees and the Editorial Policy Checklist.

## Statistics

For all statistical analyses, confirm that the following items are present in the figure legend, table legend, main text, or Methods section.

| n/a | Confirmed | |
|---|---|---|
| ☐ | ☒ | The exact sample size ($n$) for each experimental group/condition, given as a discrete number and unit of measurement |
| ☒ | ☐ | A statement on whether measurements were taken from distinct samples or whether the same sample was measured repeatedly |
| ☐ | ☒ | The statistical test(s) used AND whether they are one- or two-sided<br>*Only common tests should be described solely by name; describe more complex techniques in the Methods section.* |
| ☒ | ☐ | A description of all covariates tested |
| ☒ | ☐ | A description of any assumptions or corrections, such as tests of normality and adjustment for multiple comparisons |
| ☒ | ☐ | A full description of the statistical parameters including central tendency (e.g. means) or other basic estimates (e.g. regression coefficient) AND variation (e.g. standard deviation) or associated estimates of uncertainty (e.g. confidence intervals) |
| ☐ | ☒ | For null hypothesis testing, the test statistic (e.g. $F$, $t$, $r$) with confidence intervals, effect sizes, degrees of freedom and $P$ value noted<br>*Give P values as exact values whenever suitable.* |
| ☒ | ☐ | For Bayesian analysis, information on the choice of priors and Markov chain Monte Carlo settings |
| ☒ | ☐ | For hierarchical and complex designs, identification of the appropriate level for tests and full reporting of outcomes |
| ☒ | ☐ | Estimates of effect sizes (e.g. Cohen's $d$, Pearson's $r$), indicating how they were calculated |

*Our web collection on statistics for biologists contains articles on many of the points above.*

## Software and code

Policy information about availability of computer code

| Data collection | Geneious version 9.0.5 (Licensed, paid version used in this study, free versions available)<br>IDBA_UD version 1.1.1<br>Bowtie2 aligner version 2.3.5.1 |
|---|---|
| Data analysis | Prodigal V2.6.3<br>usearch v10.0.240_i86linux64, 1057Gb RAM, 80 cores<br>tRNAscan-SE 2.0<br>MUSCLE v3.8.31<br>BLASTp version 2.10.0+<br>BLASTn version 2.10.0+<br>hmmsearch (built in HMMER 3.3)<br>ra2.py<br>minced V0.4.2<br>MMseqs2<br>HHpred version 3.0.3<br>IQtree version 1.6.12<br>BBTools version 37.50<br>sickle version 1.33 |

For manuscripts utilizing custom algorithms or software that are central to the research but not yet described in published literature, software must be made available to editors/reviewers. We strongly encourage code deposition in a community repository (e.g. GitHub). See the Nature Research guidelines for submitting code & software for further information.

## Data

Policy information about [availability of data](availability of data)

All manuscripts must include a [data availability statement](data availability statement). This statement should provide the following information, where applicable:

- Accession codes, unique identifiers, or web links for publicly available datasets
- A list of figures that have associated raw data
- A description of any restrictions on data availability

The genomes of pmoC-phages and their relatives reported in this study have been deposited at NCBI under PRJNA645206, and are also available at Figshare (https://figshare.com/projects/pmoC-phages_in_freshwater_ecosystems/76623). The read archive and other accession information is provided in Supplementary Table 8. The pmoACB and Cas proteins HMM datasets are available at http://tigrfams.jcvi.org/cgi-bin/Listing.cgi. The 16S rRNA gene HMM database is available at https://github.com/christophertbrown/bioscripts/tree/master/databases.

# Field-specific reporting

Please select the one below that is the best fit for your research. If you are not sure, read the appropriate sections before making your selection.

☒ Life sciences          ☐ Behavioural & social sciences          ☐ Ecological, evolutionary & environmental sciences

For a reference copy of the document with all sections, see [nature.com/documents/nr-reporting-summary-flat.pdf](nature.com/documents/nr-reporting-summary-flat.pdf)

# Life sciences study design

All studies must disclose on these points even when the disclosure is negative.

| | |
|---|---|
| Sample size | Base Mine Lake (BML): N = 28 samples, each sample was collected and sequenced individually.<br>BML source (BML_S): N = 1 sample, the sample was collected and sequenced independently.<br>Lake Mendota (LM): N = 91 samples, the samples were from Lake Mendota in Wisconsin, USA in 2008-2012.<br>Crystal Bog (CB): N = 79 samples, the samples were from Crystal Bog in Wisconsin, USA in 2007-2009.<br>Lake Rotsee (LR): N = 6 samples, all with both metagenomic and metatranscriptomic datasets from Lake Rotsee of Switzerland (2017 and 2018).<br><br>We analyzed all the datasets that were collected in this study (i.e., BML and BML_S) and those reported in previous studies (i.e., LM, CB, LR). We did not select samples from them. As we are searching the existence of pmoC-phages in the corresponding samples, the sample sizes are sufficient for presence and absense analyses. |
| Data exclusions | None |
| Replication | Sample collection was not replicated. |
| Randomization | Randomization is not applicable because there were no experimental groups designated in this study. |
| Blinding | Blinding was not performed because it was not applicable to this study. This study was a survey of various populations, and was not dependent on the presence / absence of certain characteristics. |

# Reporting for specific materials, systems and methods

We require information from authors about some types of materials, experimental systems and methods used in many studies. Here, indicate whether each material, system or method listed is relevant to your study. If you are not sure if a list item applies to your research, read the appropriate section before selecting a response.

### Materials & experimental systems

| n/a | Involved in the study |
|---|---|
| ☒ ☐ | Antibodies |
| ☒ ☐ | Eukaryotic cell lines |
| ☒ ☐ | Palaeontology |
| ☒ ☐ | Animals and other organisms |
| ☒ ☐ | Human research participants |
| ☒ ☐ | Clinical data |

### Methods

| n/a | Involved in the study |
|---|---|
| ☒ ☐ | ChIP-seq |
| ☒ ☐ | Flow cytometry |
| ☒ ☐ | MRI-based neuroimaging |

