## [Peer review file · Nature Microbiology]

Peer Review Information

Journal: Nature Microbiology

Manuscript Title: Large Freshwater Phages with the Potential to Augment Aerobic Methane Oxidation

Corresponding author name(s): Jillian Banfield

Reviewer Comments & Decisions:

Decision Letter, initial version:

Dear Jill,

Thank you for your patience while your manuscript "Large Freshwater Phages with the Potential to Augment Aerobic Methane Oxidation" was under peer-review at Nature Microbiology. It has now been seen by 3 referees, whose expertise and comments you will find at the of this email. You will see from their comments below that while they find your work of interest, some important points are raised. We are very interested in the possibility of publishing your study in Nature Microbiology, but would like to consider your response to these concerns in the form of a revised manuscript before we make a final decision on publication.

In particular, you will see that referees #1 and #3 (both with expertise in phage genomics) have some comments that require clarification in the text. Referee #2, our methane oxidation referee, asks for microbiological supporting evidence to show that these phage do contribute to this biogeochemical process via isolation. However, we understand that this is difficult to do and would take a considerable amount of time, so we are willing to overrule these points. The rest referees' reports are clear and the remaining issues should be straightforward to address.

If you have not done so already please begin to revise your manuscript so that it conforms to our Article format instructions at <http://www.nature.com/nmicrobiol/info/final-submission/>

The usual length limit for a Nature Microbiology Article is six display items (figures or tables) and 3,000 words. We have some flexibility, and can allow a revised manuscript at 3,500 words, but please consider this a firm upper limit. There is a trade-off of ~250 words per display item, so if you need more space, you could move a Figure or Table to Supplementary Information.

Some reduction could be achieved by focusing any introductory material and moving it to the start of your opening 'bold' paragraph, whose function is to outline the background to your work, describe in a sentence your new observations, and explain your main conclusions. The discussion should also be limited. Methods should be described in a separate section following the discussion, we do not place a word limit on Methods.

Nature Microbiology titles should give a sense of the main new findings of a manuscript, and should not contain punctuation. Please keep in mind that we strongly discourage active verbs in titles, and that they should ideally fit within 90 characters each (including spaces).

Please include a data availability statement as a separate section after Methods but before references, under the heading "Data Availability". This section should inform readers about the availability of the data used to support the conclusions of your study. This information includes accession codes to public repositories (data banks for protein, DNA or RNA sequences, microarray, proteomics data etc...), references to source data published alongside the paper, unique identifiers such as URLs to data repository entries, or data set DOIs, and any other statement about data availability. At a minimum, you should include the following statement: "The data that support the findings of this study are available from the corresponding author upon request", mentioning any restrictions on availability. If DOIs are provided, we also strongly encourage including these in the Reference list (authors, title, publisher (repository name), identifier, year). For more guidance on how to write this section please see: <http://www.nature.com/authors/policies/data/data-availability-statements-data-citations.pdf>

To improve the accessibility of your paper to readers from other research areas, please pay particular attention to

the wording of the paper's opening bold paragraph, which serves both as an introduction and as a brief, non-technical summary in about 150 words. If, however, you require one or two extra sentences to explain your work clearly, please include them even if the paragraph is over-length as a result. The opening paragraph should not contain references. Because scientists from other sub-disciplines will be interested in your results and their implications, it is important to explain essential but specialised terms concisely. We suggest you show your summary paragraph to colleagues in other fields to uncover any problematic concepts.

If your paper is accepted for publication, we will edit your display items electronically so they conform to our house style and will reproduce clearly in print. If necessary, we will re-size figures to fit single or double column width. If your figures contain several parts, the parts should form a neat rectangle when assembled. Choosing the right electronic format at this stage will speed up the processing of your paper and give the best possible results in print. We would like the figures to be supplied as vector files - EPS, PDF, AI or postscript (PS) file formats (not raster or bitmap files), preferably generated with vector-graphics software (Adobe Illustrator for example). Please try to ensure that all figures are non-flattened and fully editable. All images should be at least 300 dpi resolution (when figures are scaled to approximately the size that they are to be printed at) and in RGB colour format. Please do not submit Jpeg or flattened TIFF files. Please see also 'Guidelines for Electronic Submission of Figures' at the end of this letter for further detail.

Figure legends must provide a brief description of the figure and the symbols used, within 350 words, including definitions of any error bars employed in the figures.

Please include a statement before the acknowledgements naming the author to whom correspondence and requests for materials should be addressed.

Finally, we require authors to include a statement of their individual contributions to the paper -- such as experimental work, project planning, data analysis, etc. -- immediately after the acknowledgements. The statement should be short, and refer to authors by their initials. For details please see the Authorship section of our joint Editorial policies at http://www.nature.com/authors/editorial_policies/authorship.html

* include a point-by-point response to any editorial suggestions and to our referees. Please include your response to the editorial suggestions in your cover letter, and please upload your response to the referees as a separate document.

* ensure it complies with our format requirements for Letters as set out in our guide to authors at www.nature.com/nmicrobiol/info/gta/

* state in a cover note the length of the text, methods and legends; the number of references; number and estimated final size of figures and tables

* resubmit electronically if possible using the link below to access your home page:

{REDACTED}

*This url links to your confidential homepage and associated information about manuscripts you may have submitted or be reviewing for us. If you wish to forward this e-mail to co-authors, please delete this link to your homepage first.

Please ensure that all correspondence is marked with your Nature Microbiology reference number in the subject line.

Nature Microbiology is committed to improving transparency in authorship. As part of our efforts in this direction, we are now requesting that all authors identified as 'corresponding author' on published papers create and link their Open Researcher and Contributor Identifier (ORCID) with their account on the Manuscript Tracking System (MTS), prior to acceptance. This applies to primary research papers only. ORCID helps the scientific community achieve unambiguous attribution of all scholarly contributions. You can create and link your ORCID from the home page of the MTS by clicking on 'Modify my Springer Nature account'. For more information please visit please visit <http://www.springernature.com/orcid>.

We hope to receive your revised paper within six weeks. If you cannot send it within this time, please let us know.

Reviewer Expertise:

Referee #1: phage genomics, bioinformatics

Referee #2: environmental microbial communities, methane oxidation

Referee #3: phage genomics

Reviewers Comments:

Reviewer #1 (Remarks to the Author):

Large Freshwater Phages with the Potential to Augment Aerobic Methane Oxidation

This paper by Chen and colleagues describes the identification of a set of interesting, related, phages from freshwater environments that encode one particular gene, *pmoC*, whose product is involved in methane oxidation.

The paper is a delight to read, presents novel results, identifies an interesting conundrum, and speculates on solutions, and I recommend it for publication in Nature Microbiology.

Chen et al, identified a set of phages among their metagenome-assembled genomes, and from that dataset identified phages that contain the *pmoC* gene. After manual curation, they were able to generate 18 complete phage genomes and four additional phage genomes.

Perhaps the most interesting observation is that they only found phages with *pmoC* and not with the other two genes typically required for methane oxidation, *pmoA* and *pmoB*. As was demonstrated previously by the Lidstrom group (ref 15), the *PmoC* proteins appear divergent at the N- and C-terminals. It is interesting to speculate whether that divergence is associated with the function of the proteins to drive methane oxidation.

A question that occurred repeatedly through reading the manuscript is "what is the role of *PmoC* in phage replication?". However, I realize that the role of *PmoC* in methane oxidation is not entirely clear, especially the role of the additional copies of the gene in the genomes that are not associated with *pmoAB*. Again, it is thought-provoking to suggest that *PmoC* may be rate-limiting in methane oxidation and thus the phage copy (and perhaps additional bacterial copies) are increasing the rate or longevity of this reaction. However, the role of *PmoC* for the phage may not have anything to do with methane oxidation directly, it may be a way for the phage to increase intracellular concentrations of NAD^+ that are required for replication.

Perhaps surprisingly the phages were identified from microbial metagenomes that had been filtered through 0.22 μm filters. Typically these filters do not capture phages, and so I wonder if these phages were discovered because either they were undergoing replication during filtering (e.g. the act of filtering induced the phages to

enter the lytic cycle), because they are larger and thus more likely to be retained by the filters, or perhaps many microbial metagenomics datasets have phages in them that others have not explored?

The close relationship between the pmoC-encoding phages and the cyanophages, and the observation that these phages can infect both Gamma-proteobacteria (Methyloparacoccus and Methylobacter) and Alpha-proteobacteria (Methylocystis) lends me to wonder whether there is any data suggesting that they may also be able to infect cyanobacteria.

Minor comments:

Lines 182-189: it would help the reader to provide a brief (1-2 sentence) summary of the role of HSP20 in the cell.

In figure 2, the red * is not quite clear in its relationship to the pull-out for CB pmoC's. I would suggest labeling it as "CB Methylocystis copy 3" to make it clearer.

Reviewer #2 (Remarks to the Author):

This is an interesting study looking at freshwater phage that carry a single copy of pmoC, encoding part of the particulate methane monooxygenase complex that is responsible for biological methane oxidation. These phage have large genomes and the pmoC gene they carry has a high degree of similarity to pmoC from methanotrophs known to inhabit the same environments which the phage genomes were isolated from. The authors suggest that because the phage-associated PmoC has a high degree of similarity and are phylogenetically-related to these co-existing methanotrophs and relative abundances of each in some environments correlate then this phage PmoC has similar functions to PmoC encoded by the singleton copies of pmoC in methanotrophs. They also suggest from sequencing data that the most rapidly growing methanotrophs they observe (via DNA and RNA sequences) was infected by pmoC-containing phages and that this infection could modulate the cycling of methane by these methanotrophs. Although an excellent meta'omics study, as far as I can see, the data are largely correlative and suggest a number of interesting things, but what the study is really lacking is some microbiology. All of the genera of methanotrophs that are mentioned have been cultivate in various labs around the world and the necessary methodology is available to do this. Probably the one to aim for is Methyloparacoccus although Methylocystis and Methylobacter strains should also be relatively straightforward to isolate. The manuscript could have been strengthened considerably by doing some isolation work and showing which methanotrophs were infected and indeed if there is a close link between these phage and methane oxidation by these infected methanotrophs. Once the isolates are available then the hypotheses generated can be tested. There is also literature on phage for methanotrophs and their isolation, which will provide methods for isolating the phage once a range of hosts have been obtained. The manuscript contains too many inferences, and is full of phrases such as "suggesting that", "potentially significant", "could have been", " could sustain," "may be a key player" etc. As the manuscript stands at the moment, a very nice 'omics study but the conclusions drawn are far too speculative and there are too many inferences to make this work a significant advance in the field.

L 83 why not isolate these potentially key methanotrophs and test them in the lab?

92 how can you tell what the growth rate is if you don't have a culture?

95 low temperature won't inhibit" growth (it might slow it down, it might not-this could be tested with isolates).

114-116 this phrase is very speculative

130 I am not sure that the active site of pMMO has been defined at this level of detail yet? This is very speculative.

167 predicting the hosts from molecular data is good but then this hypothesis needs testing

202 high in situ activities of 'phage? This doesn't make sense?

208 I am not convinced that the data really show that expression of pmoC is important during the late phase of phage replication.

230-239 this is highly speculative discussion

250-251 again this is a very speculative statement.

Reviewer #3 (Remarks to the Author):

This paper describes a novel set of bacteriophages associated with methane oxidation. It builds on the body of evidence that is revealing how bacteriophages contribute to microbial metabolism. The first time it highlights the presence, and potential importance of *pmoC* in methane oxidation. The authors examined a man-made lake and two natural lakes. Phylogenetic analysis showed that the gene is closely related to its bacterial ancestors.

The paper is well written, represents a significant body of novel and interesting work, carried out to a high standard and I believe will be of interest to a wide readership. The work reanalysed previously curated metagenomic data with the view to identify phages, specifically those involved in methane metabolism to methanol. They managed to identify the scaffolds of 22 phages, 18 of which were complete. They also looked at transcriptomic data in one of the lakes to try to ascertain expression profiles. My recommendation would be that this work is published in Nature Microbiology subject to minor revision, specific comments are below.

The abstract captures the essence of the idea and is clearly written. Again the introduction sets the scene within the context of both methane oxidation and auxiliary metabolic genes in bacteriophages.

The authors start by detailing the geobiology of the lake systems, highlighting the very high levels of methane present. They then go on to show that the main microbial host is *Methyloparacoccus_57*.

The authors then search through many other datasets available to them to look for phage encoded *pmoC* and found that the similar in the central membrane- and periplasmic associated portions, but divergent at the cytoplasmic N- and C-termini. They showed that the phage encoded version was most similar to the surrounding bacteria, compared to sequences from other lakes. They also showed that the abundance of the phage encoded gene was often higher than that of the bacterial version. The authors struggled to make an exact host prediction, essentially it is made on correlation of phages with the host and on the similarity of the gene in question, *pmoC*. Although this is not ideal I don't really see what the authors can do differently at the moment other than ultimately isolate them.

The authors tried to see which other genes might be present alongside *pmoC* and determine if they also contributed to metabolism. Unfortunately the system is very messy and they didn't really make clear findings. Again I feel they've done what can be done and it doesn't really detract from the main point of the paper. They then show that there is a high expression of *pmoC* genes in one of their lakes they examine. The discussion picks up on the threads and reasons why such work has previously been overlooked and also on why only one of the genes involved in methane metabolism is present in the bacteriophages. They also discussed the potential impacts of this phage. The figures are clearly presented and well explained.

In summary, although there is clearly a lot of work to still be done I feel that this paper is really interesting, and takes the field of phage encoded metabolic genes to an interesting and new level.

Comments for improvement

There seems to be a slight jump in narrative between line 116 and 117 - the authors previously discussed the main bacterial host but don't say why they then link this information to the phage *pmoC*. It would be good if they could link this more clearly.

More could be made of the actual phages themselves -clearly the focus is on *pmoC* genes that the authors do state a very high number of tRNA's - so it would be interesting if they stated that correlated to codon usage.

Line 207 the authors state that they interpret the high levels of expression of *pmoC*, alongside high levels of expression of phage structural genes to mean that the *pmoC* gene is expressed late on in the phage cycle. I am unclear about how they make this interpretation as couldn't it be that as the phages are very abundant, they are being expressed that the middle stage of the phage lifestyle, or even early which is when you would expect them to be expressed. Perhaps the authors could clarify this.

Author Rebuttal to Initial comments**Reviewer #1 (Remarks to the Author):****Large Freshwater Phages with the Potential to Augment Aerobic Methane Oxidation**

This paper by Chen and colleagues describes the identification of a set of interesting, related, phages from freshwater environments that encode one particular gene, *pmoC*, whose product is involved in methane oxidation.

The paper is a delight to read, presents novel results, identifies an interesting conundrum, and speculates on solutions, and I recommend it for publication in Nature Microbiology.

Chen et al, identified a set of phages among their metagenome-assembled genomes, and from that dataset identified phages that contain the *pmoC* gene. After manual curation, they were able to generate 18 complete phage genomes and four additional phage genomes.

Perhaps the most interesting observation is that they only found phages with *pmoC* and not with the other two genes typically required for methane oxidation, *pmoA* and *pmoB*. As was demonstrated previously by the Lidstrom group (ref 15), the *PmoC* proteins appear divergent at the N- and C-terminals. It is interesting to speculate whether that divergence is associated with the function of the proteins to drive methane oxidation.

Response: We thank the reviewer for their careful reading and overall positive review of the manuscript.

A question that occurred repeatedly through reading the manuscript is “what is the role of *PmoC* in phage replication?”. However, I realize that the role of *PmoC* in methane oxidation is not entirely clear, especially the role of the additional copies of the gene in the genomes that are not associated with *pmoAB*. Again, it is thought-provoking to suggest that *PmoC* may be rate-limiting in methane oxidation and thus the phage copy (and perhaps additional bacterial copies) are increasing the rate or longevity of this reaction. However, the role of *PmoC* for the phage may not have anything to do with methane oxidation directly, it may be a way for the phage to increase intracellular concentrations of NAD^+ that are required for replication.

Response: The reviewer raises an interesting point regarding how NAD^+ generated in the conversion of methane to methanol may be useful from the perspective of increasing the phage replication rate. We have added this comment to the manuscript, along with this citation (<https://www.ncbi.nlm.nih.gov/pmc/articles/PMC5388814/>). However, we do not state that this outcome has nothing to do with methane oxidation directly, as methane is oxidized in the reaction. We apologize if we misunderstood the reviewer’s comment.

Perhaps surprisingly the phages were identified from microbial metagenomes that had been filtered through $0.22\mu\text{m}$ filters. Typically these filters do not capture phages, and so I wonder if these phages were discovered because either they were undergoing replication during filtering (e.g. the act of filtering induced the phages to enter the lytic cycle), because they are larger and thus more likely to be retained by the filters, or perhaps many microbial metagenomics datasets have phages in them that others have not explored?

Response: A prior study from our lab and collaborators showed that large phages are widely distributed across ecosystems and are routinely recovered from water samples collected by filtering. We anticipate that the phages detected in our study are likely $< 0.2\mu\text{m}$ diameter, but it is our understanding that objects that are smaller than the pore size of filters can be trapped on them. We also consider it likely that large phages have been missed in many microbial metagenomic analyses, either because phages are not of interest to the investigators or, more likely, their genomes are too fragmented to be recognized. Certainly, some large phages may have been replicating when the samples were collected. In fact, we reported the expression of genes related to late replication in the current study.

The close relationship between the *pmoC*-encoding phages and the cyanophages, and the observation that these phages can infect both Gamma-proteobacteria (*Methyloparacoccus* and *Methylobacter*) and Alpha-proteobacteria (*Methylocystis*) lends me to wonder whether there is any data suggesting that they may also be able to infect cyanobacteria.

Response: The reviewer raises an interesting question. However, we do not have data (e.g., CRISPR-Cas targeting from Cyanobacteria) to support this possibility.

Minor comments:

Lines 182-189: it would help the reader to provide a brief (1-2 sentence) summary of the role of HSP20 in the cell.

Response: As we note, HSP20 is a core gene in Cyanophages and provide this citation (<https://www.ncbi.nlm.nih.gov/pmc/articles/PMC3037559/>), which states: "Other cyanophage core genes include proteins that likely encode basic phage functions, such as a heat shock family protein (hsp20) that might be important for scaffolding during maturation of the capsid".

We have added a statement: HSP20 is a small heat shock protein that may improve the survival of the host bacteria when they are challenged by elevated temperature, although it also has been suggested HSP20 might be important for scaffolding during maturation of the capsid.

In figure 2, the red * is not quite clear in its relationship to the pull-out for CB pmoC's. I would suggest labeling it as "CB Methylocystis copy 3" to make it clearer.

Response: Done as suggested.

Reviewer #2 (Remarks to the Author):

This is an interesting study looking at freshwater phage that carry a single copy of pmoC, encoding part of the particulate methane monooxygenase complex that is responsible for biological methane oxidation. These phage have large genomes and the pmoC gene they carry has a high degree of similarity to pmoC from methanotrophs known to inhabit the same environments which the phage genomes were isolated from. The authors suggest that because the phage-associated PmoC has a high degree of similarity and are phylogenetically-related to these co-existing methanotrophs and relative abundances of each in some environments correlate then this phage PmoC has similar functions to PmoC encoded by the singleton copies of pmoC in methanotrophs. They also suggest from sequencing data that the most rapidly growing methanotrophs they observe (via DNA and RNA sequences) was infected by pmoC-containing phages and that this infection could modulate the cycling of methane by these methanotrophs.

Response: We thank the reviewer for their careful reading of this manuscript.

Although an excellent meta'omics study, as far as I can see, the data are largely correlative and suggest a number of interesting things, but what the study is really lacking is some microbiology. All of the genera of methanotrophs that are mentioned have been cultivate in various labs around the world and the necessary methodology is available to do this. Probably the one to aim for is Methyloparacoccus although Methylocystis and Methylobacter strains should also be relatively straightforward to isolate. The manuscript could have been strengthened considerably by doing some isolation work and showing which methanotrophs were infected and indeed if there is a close link between these phage and methane oxidation by these infected methanotrophs. Once the isolates are available then the hypotheses generated can be tested. There is also literature on phage for methanotrophs and their isolation, which will provide methods for isolating the phage once a range of hosts have been obtained. The manuscript contains too many inferences, and is full of phrases such as "suggesting that", "potentially significant", "could have been", " could sustain," "may be a key player" etc. As the manuscript stands at the moment, a very nice 'omics study but the conclusions drawn are far too speculative and there are too many inferences to make this work a significant advance in the field.

Response: We thank the reviewer for this suggestion regarding isolation and experiments. We agree that these bacteria likely could be isolated (if original samples were available) but the isolation of large phages is notoriously difficult and only on the order of ~100 that are > 200 kbp in length have been isolated to date, to our knowledge. We hope that the current study will encourage others to conduct the detailed experiments that the current study motivates.

L83 why not isolate these potentially key methanotrophs and test them in the lab?

Response: Although we agree that these experiments would be of interest, they are beyond the scope of the current study.

L92 how can you tell what the growth rate is if you don't have a culture?

Response: Culture is not the only way to evaluate the replication rate of a microorganism. It has long been known that bacteria replicate their genomes from the origin to terminus, so replication results in systematic decrease in coverage from the origin to the terminus. There have been several studies that have leveraged data from culture experiments to calibrate the genome coverage trends. Notably, the iRep tool makes quantification of replication rates for bacteria possible from draft genomes, so long as coverage and bin quality thresholds are exceeded. This is already described in the methods, so we have not made any change to the manuscript.

L95 low temperature won't inhibit" growth (it might slow it down, it might not-this could be tested with isolates).

Response: Thanks for this comment. We modify "inhibited" to "slowed down" in the revised manuscript.

L114-116 this phrase is very speculative

Response: The reviewer refers to:

"

113 BML_3 (64% and 75% of genomes aligned, respectively). Our reanalysis of published $^{13}\text{CH}_4$ -based DNA-SIP (stable
114 isotope probing) data ²⁶ detected PDSYNTPWS_1 in the heavy DNA-SIP fraction. This fraction was dominated by
115 Methyloparacoccus_57, suggesting that this bacterium oxidized ^{13}C -enriched methane and could have been the host
116 for the phage PDSYNTPWS_1.

"

We agree that this is speculative in so far as we found a host bacterium and associated PmoC-phage in a sample for which there is isotope data indicating biological methane oxidation. We have revised the sentence to read:

This fraction was dominated by Methyloparacoccus_57. Based on the co-occurrence of the host and phage in a sample in which biological methane oxidation was demonstrated isotopically, we suggest that Mehtyloparacoccus_57 may have been the host for phage PDSYNPWS_1. Also supporting this association is the high genomic and phylogenetic similarity between PDSYNPWS_1 and BML_3 (Fig. 3), whose host was predicted as Mehtyloparacoccus_57.

L130 I am not sure that the active site of pMMO has been defined at this level of detail yet? This is very speculative.

Response: In the manuscript, we report what is known about key residues required from PmoC function:

"

122 We confirmed the high similarity of the bacterial and phage-associated PmoC predicted from all datasets to
123 PmoC of previously described alphaproteobacterial and gammaproteobacterial methanotrophs (Supplementary
124 Tables 4 and 5). Alignment of these PmoC with references from well-known bacterial methanotrophs ²⁹ confirmed the
125 presence of the residues necessary for the copper-binding site, i.e., Asp¹⁵⁶, His¹⁶⁰, and His¹⁷³ (Fig. 2a, Supplementary
126 Fig. 9) and required for O₂ binding and methane oxidation ^{12,30}. Interestingly, the bacterial and phage-associated PmoC
127 sequences were generally very similar in the central membrane- and periplasmic associated portions, but divergent at
128 the cytoplasmic N- and C-termini. The *pmoC* genes in four of the pmoC-phages were fragmented into two pieces and
129 another one (LM_8) contained only the C-terminus (Supplementary Fig. 10). In addition, the *pmoC* gene from CB_5
130 exhibited within-population variation as a subset of phages lacked the central region where the active site is located
131 (Supplementary Fig. 10).

"

The statement that the alignment confirmed the presence of these residues is not speculative, nor are the sequence

similarity patterns reported or the gene fragmentation patterns. Thus, we are uncertain how to address this reviewer's comment.

L167 predicting the hosts from molecular data is good but then this hypothesis needs testing

Response: We concur that experiments will be important for further understanding. The value of the current analysis is that it makes a strong prediction regarding how future experiments should be designed. And to our knowledge, the sequence similarity of AMGs is widely used to predict host/phage relationship (per the literature we cited).

L202 high in situ activities of 'phage? This doesn't make sense?

Response: We are wondering if the reviewer does not like the idea of phages having *in situ* activity? (as opposed to activities of the host bacterium). The data reported relating to transcriptional data indicative of phage replication at the time of sampling. We have revised the text to read:

“Of the 6 *pmoC*-phages from Lake Rotsee, LR_4, LR_5 and LR_6 (genome sizes > 300 kbp; Table 1) showed high transcriptional levels indicative of replication at the time of sampling in November and December of 2017. Transcript data indicate that only LR_4 was highly active in the January 2018 samples (Fig. 5a, Supplementary Fig. 23).”

L208 I am not convinced that the data really show that expression of *pmoC* is important during the late phase of phage replication.

Response: We have clarified that, as the expression of structural proteins is typically associated with the late phase of replication, co-expression of *pmoC* with those genes suggest that *pmoC* is important during the late phase of phage replication. We have revised the text to read:

“Interestingly, the *pmoC* genes of LR_4, LR_5 and LR_6 were highly expressed (generally among the top 20 most active genes), as were genes encoding for phage DNA packaging and particle assembly related proteins, including major capsid, prohead, phage tail, tail fiber, tail sheath and scaffolding proteins (Figs. 5b-d). Given that structural genes are generally expressed late in replication, we interpret the co-expression pattern to indicate that *pmoC* is important during the late phase of phage replication.”

L230-239 this is highly speculative discussion.

Response: We assume that it is the statement that “additional *pmoC* genes, either in the bacterial methanotroph or phage genome (Supplementary Table 3), could sustain methane oxidation under abnormal conditions “ is the part that the reviewer considers to be speculative. We agree that this is a prediction, but it seems logical that there is a reason for the gene to be consistently found in the genomes of methanotrophs. In response to this comment, we have revised the statement to read:

“Given that the structure of PmoC is largely disordered when the cell membrane is perturbed⁴², we suggest that the additional *pmoC* genes, either in the bacterial methanotroph or phage genome (Supplementary Table 3), could augment methane oxidation. Although we do not have data to constrain how this occurs, it seems reasonable to speculate that it may sustain methane oxidation under abnormal environmental conditions. The availability of an alternative enzyme also may be beneficial when metals used in the normal bacterial subunit are in low abundance, given that Zn or Cu can be used in the PmoC catalytic site.”

L250-251 again this is a very speculative statement.

Response: A decline in methanotroph abundance should reduce methane oxidation rates. The observation is that the bacterial host declined after the phages appeared in sufficient abundance to be detected, so it seems to us a reasonable deduction that the decline was linked to phage-induced lysis. The statement is already flagged as a suggestion so we have not changed the manuscript in response to this comment.

Reviewer #3 (Remarks to the Author):

This paper describes a novel set of bacteriophages associated with methane oxidation. It builds on the body of evidence that is revealing how bacteriophages contribute to microbial metabolism. The first time it highlights the presence, and potential importance of pmoC in methane oxidation. The authors examined a man-made lake and two natural lakes. Phylogenetic analysis showed that the gene is closely related to its bacterial ancestors.

The paper is well written, represents a significant body of novel and interesting work, carried out to a high standard and I believe will be of interest to a wide readership. The work reanalysed previously curated metagenomic data with the view to identify phages, specifically those involved in methane metabolism to methanol. They managed to identify the scaffolds of 22 phages, 18 of which were complete. They also looked at transcriptomic data in one of the lakes to try to ascertain expression profiles. My recommendation would be that this work is published in Nature Microbiology subject to minor revision, specific comments are below.

Response: We thank this reviewer for their careful reading and positive evaluation. We make one small note, however. We did not simply “identify” complete phage genomes - the genomes were manually curated to completion. This step is very rarely undertaken in genome-resolved metagenomic analyses, but is key to quality control and thus confidence in the results.

The abstract captures the essence of the idea and is clearly written. Again the introduction sets the scene within the context of both methane oxidation and auxiliary metabolic genes in bacteriophages.

Response: Thank you!

The authors start by detailing the geobiology of the lake systems, highlighting the very high levels of methane present. They then go on to show that the main microbial host is *Methyloparacoccus_57*.

The authors then search through many other datasets available to them to look for phage encoded pmoC and found that the similar in the central membrane- and periplasmic associated portions, but divergent at the cytoplasmic N- and C-termini. They showed that the phage encoded version was most similar to the surrounding bacteria, compared to sequences from other lakes. They also showed that the abundance of the phage encoded gene was often higher than that of the bacterial version. The authors struggled to make an exact host prediction, essentially it is made on correlation of phages with the host and on the similarity of the gene in question, pmoC. Although this is not ideal I don't really see what the authors can do differently at the moment other than ultimately isolate them.

The authors tried to see which other genes might be present alongside pmoC and determine if they also contributed to metabolism. Unfortunately the system is very messy and they didn't really make clear findings. Again I feel they've done what can be done and it doesn't really detract from the main point of the paper. They then show that there is a high expression of pmoC genes in one of their lakes they examine.

The discussion picks up on the threads and reasons why such work has previously been overlooked and also on why only one of the genes involved in methane metabolism is present in the bacteriophages. They also discussed the potential impacts of this phage. The figures are clearly presented and well explained.

In summary, although there is clearly a lot of work to still be done I feel that this paper is really interesting, and takes the field of phage encoded metabolic genes to an interesting and new level.

Response: We thank the reviewer for their careful reading and overall positive review of the manuscript.

Comments for improvement:

There seems to be a slight jump in narrative between line 116 and 117 - the authors previously discussed the main bacterial host but don't say why they then link this information to the phage pmoC. It would be good if they could link this more clearly.

109 Reanalysis of the published oil sands datasets (Supplementary Information) detected one *pmoC*-phage scaffold
110 (TP6_1) in a Suncor tailings pond sampled in 2012²⁷. Additionally, phages similar to TP6_1 and BML_3 were detected
111 in two other samples from Alberta, i.e., TP_MLSB collected in 2011²⁷ and PDSYNTPWS collected in 2012²⁶. From
112 PDSYNTPWS, we curated a phage genome without *pmoC* (referred to as “PDSYNTPWS_1”), which is 99% similar to
113 BML_3 (64% and 75% of genomes aligned, respectively). Our reanalysis of published ¹³CH₄-based DNA-SIP (stable
114 isotope probing) data²⁶ detected PDSYNTPWS_1 in the heavy DNA-SIP fraction. This fraction was dominated by
115 *Methyloparacoccus_57*, suggesting that this bacterium oxidized ¹³C-enriched methane and could have been the host
116 for the phage PDSYNTPWS_1.

117 To test for phage-associated *pmoC* in other lakes reported to emit methane¹⁷, we searched our previously
118 published metagenomic datasets from Lake Mendota (LM) and Crystal Bog (CB) in Wisconsin (USA)¹⁹, and those
119 recently published from Lake Rotsee (LR)¹⁶. The LM, CB and LR datasets were reanalyzed (Methods), and HMM-based
120 searches detected *pmoC* on phage scaffolds from all the three lakes (Supplementary Table 5), suggesting the
121 potentially wide distribution of related phages in habitats with methane.

Response: We concur that there is a switch in the topic at this point. In the paragraph that ends at line 116, we describe the identification of *pmoC*-phages in published oil sands datasets and infer the host of *pmoC*-phage PDSYNTPWS_1. In the following paragraph, we move on to look for phage with *PmoC* in similar environments. It isn't clear to us how we might better frame this transition, given that in the second of these paragraphs we state that we went on “To test for phage-associated *pmoC* in other lakes reported to emit methane...”.

More could be made of the actual phages themselves - clearly the focus is on *pmoC* genes that the authors do state a very high number of tRNA's - so it would be interesting if they stated that correlated to codon usage.

Response: We thank this reviewer for this suggestion. Motivated by this comment we compared the tRNAs of phages with the inventory of tRNAs encoded in these phage genomes and found statistically supported similarity. This information has been added to the manuscript, and the following figures added to the Supplementary Information.

Figure legend: **The codon usage frequency of pmoC-phages and their relatives.** Upper panel: Clustering analyses of phages based on codon usage frequency. Bottom panel: Comparison of usage frequency of codons without tRNA encoded and with tRNA encoded in the phages. The ones with tRNA encoded by phages showed an overall higher usage frequency at a 95% confidence level (unpaired student's t-test). Each point represents the usage frequency of a given codon in a given phage.

Line 207 the authors state that they interpret the high levels of expression of pmoC, alongside high

levels of expression of phage structural genes to mean that the *pmoC* gene is expressed late on in the phage cycle. I am unclear about how they make this interpretation as couldn't it be that as the phages are very abundant, they are being expressed that the middle stage of the phage lifestyle, or even early which is when you would expect them to be expressed. Perhaps the authors could clarify this.

Response: We interpret this comment to suggest that the phages may be in various stages of their replication cycle, which we concur is very possible. However, it would seem that for us to detect high levels of both structural proteins and *pmoC* transcripts that these transcripts were likely being co-expressed by phage during the late portion of their replication cycle (please also see above for our response to a similar comment from Reviewer #2). We have modified the statement to indicate the nature of this inference.

Decision Letter, first revision:

Dear Jill,

Thank you for your patience while your manuscript "Large Freshwater Phages with the Potential to Augment Aerobic Methane Oxidation" was under peer review at Nature Microbiology. It has now been seen by our referees, and in the light of their advice I am delighted to say that we can in principle offer to publish it. First, however, we would like you to revise your paper to address the points made by the reviewers, and to ensure that it is in Nature Microbiology format.

The referees' have no remaining comments. Editorially, we will need you to make some changes so that the paper complies with our Guide to Authors at <http://www.nature.com/nmicrobiol/info/gta>.

Nature Microbiology offers a transparent peer review option for new original research manuscripts submitted from 1st December 2019. We encourage increased transparency in peer review by publishing the reviewer comments, author rebuttal letters and editorial decision letters if the authors agree. Such peer review material is made available as a supplementary peer review file. **Please state in the cover letter 'I wish to participate in transparent peer review' if you want to opt in, or 'I do not wish to participate in transparent peer review' if you don't.** Failure to state your preference will result in delays in accepting your manuscript for publication.

Specific points:

In particular, while checking through the manuscript and associated files, we noticed the following specific points which we will need you to address:

1. Extended data. Per journal guidelines, we use "Extended Data". Please see below for additional information on how to format and refer to Extended Data. Extended Data figures should be multi-panel, similar to main figures, be cited in the main text and be directly relevant to the main results. Please select the most relevant figures for Extended Data.

2. Supplementary Information. Please move any methods related text currently in the SI into the methods section where there is no word limit.
3. Priority claims. Per journal guidelines, we recommend that you avoid the use of terms like 'new', 'novel' and other priority claims throughout the text in order to avoid any perception of grandstanding and so that the reader can focus on the significance, rather than the novelty, of the findings. Therefore, please revise lines 41 and 626 ('newly') and any other relevant sections accordingly.
4. Abstract. Please mention key bacterial names in the abstract and clarify that future work will be needed to confirm that phage pmoC does actually play a role in methane oxidation.
5. Figure size. Please ensure that figures fit comfortably onto one A4 page surrounded by white space and including the legend. Text should be at least 5 pt. Currently many figures are too small and the text is illegible. Please reformat and ensure that all text and symbols are clear.
6. Figures. While carefully checking the figures, we noted a few things that need to be revised so that the findings are clear to the reader:
 - Figure ED7 - please explain in the legend what the red and blue font represent.
 - Figure S3 and S11 - please explain in the legend what the bold font and red font represents. There are two S3 figures. Please correct.
 - Figure S12 - please indicate in the legend what the red dashed line represents.
 - Figure S14 - please provide a colour key for (b).
7. Source data. Please provide source data for all figures where appropriate. Please see below for more information on how to provide this information.
8. Data availability. Please include a data availability section at the end of the methods - see below for additional details on how to format this section. Please ensure that all accession codes are live. Please also add accession codes for any previously published datasets used in the analyses.
9. Code Availability. Please provide a Code Availability statement and deposit any custom code to GitHub.
10. Reporting checklist. Note that a final version of the reporting checklist will be published with your manuscript. Therefore, please revise this document according to the instructions found in the annotated PDF attached to this message. Please also address the points listed in the attached Word Document and make the requested changes in the manuscript.

General points:

Please read carefully through all of the following general formatting points when preparing the final version of your manuscript, as submitting the manuscript files in the required format will greatly speed the process to final acceptance of your work.

We estimate that your manuscript currently exceeds our normal length limit for Articles of about 3,000 words. We have some flexibility, and can allow a revised manuscript at 3,500 words, but please consider this a firm upper limit. You could achieve some shortening by moving some details to the Methods section that should follow the main text (the length of the Methods section is unlimited and

does not count towards the main text length).

Titles should give an idea of the main finding of the paper and ideally not exceed 90 characters (including spaces). We discourage the use of active verbs and do not allow punctuation.

The paper's summary paragraph (about 150-200 words; no references) should serve both as a general introduction to the topic, and as a brief, non-technical summary of your main results and their implications. It should start by outlining the background to your work (why the topic is important) and the main question you have addressed (the specific problem that initiated your research), before going on to describe your new observations, main conclusions and their general implications. Because we hope that scientists across the wider microbiology community will be interested in your work, the first paragraph should be as accessible as possible, explaining essential but specialised terms concisely. We suggest you show your summary paragraph to colleagues in other fields to uncover any problematic concepts.

Please include a data availability statement as a separate section after Methods but before references, under the heading "Data Availability". This section should inform readers about the availability of the data used to support the conclusions of your study. This information includes accession codes to public repositories (data banks for protein, DNA or RNA sequences, microarray, proteomics data etc...), references to source data published alongside the paper, unique identifiers such as URLs to data repository entries, or data set DOIs, and any other statement about data availability. At a minimum, you should include the following statement: "The data that support the findings of this study are available from the corresponding author upon request", mentioning any restrictions on availability. If DOIs are provided, we also strongly encourage including these in the Reference list (authors, title, publisher (repository name), identifier, year). For more guidance on how to write this section please see:

<http://www.nature.com/authors/policies/data/data-availability-statements-data-citations.pdf>

Please supply the figures as vector files - EPS, PDF, AI or postscript (PS) file formats (not raster or bitmap files), preferably generated with vector-graphics software (Adobe Illustrator for example). Try to ensure that all figures are non-flattened and fully editable. All images should be at least 300 dpi resolution (when figures are scaled to approximately the size that they are to be printed at) and in RGB colour format. Please do not submit Jpeg or flattened TIFF files. Please see also 'Guidelines for Electronic Submission of Figures' at the end of this letter for further detail.

Please view http://www.nature.com/authors/editorial_policies/image.html for more detailed guidelines.

We will edit your figures/tables electronically so they conform to Nature Microbiology style. If necessary, we will re-size figures to fit single or double column width. If your figures contain several parts, the parts should be labelled lower case a, b, and so on, and form a neat rectangle when assembled.

Please check the PDF of the whole paper and figures (on our manuscript tracking system) VERY

CAREFULLY when you submit the revised manuscript. This will be used as the 'reference copy' to make sure no details (such as Greek letters or symbols) have gone missing during file-transfer/conversion and re-drawing.

All Supplementary Information must be submitted in accordance with the instructions in the attached Inventory of Supporting Information, and should fit into one of three categories:

1. **EXTENDED DATA:** Extended Data are an integral part of the paper and only data that directly contribute to the main message should be presented. These figures will be integrated into the full-text HTML version of your paper and will be appended to the online PDF. There is a limit of 10 Extended Data figures, and each must be referred to in the main text. Each Extended Data figure should be of the same quality as the main figures, and should be supplied at a size that will allow both the figure and legend to be presented on a single legal-sized page. Each figure should be submitted as an individual .jpg, .tif or .eps file with a maximum size of 10 MB each. All Extended Data figure legends must be provided in the attached Inventory of Accessory Information, not in the figure files themselves.

2. **SUPPLEMENTARY INFORMATION:** Supplementary Information is material that is essential background to the study but which is not practical to include in the printed version of the paper (for example, video files, large data sets and calculations). Each item must be referred to in the main manuscript and detailed in the attached Inventory of Accessory Information. Tables containing large data sets should be in Excel format, with the table number and title included within the body of the table. All textual information and any additional Supplementary Figures (which should be presented with the legends directly below each figure) should be provided as a single, combined PDF. Please note that we cannot accept resupplies of Supplementary Information after the paper has been formally accepted unless there has been a critical scientific error.

All Extended Data must be called out in your manuscript and cited as Extended Data 1, Extended Data 2, etc. Additional Supplementary Figures (if permitted) and other items are not required to be called out in your manuscript text, but should be numerically numbered, starting at one, as Supplementary Figure 1, not SI1, etc.

3. **SOURCE DATA:** We strongly encourage you to provide source data for your figures whenever possible. Full-length, unprocessed gels and blots must be provided as source data for any relevant figures, and should be provided as individual PDF files for each figure containing all supporting blots and/or gels with the linked figure noted directly in the file. Numerical source data that underlie graphs are required for in vivo experiments and strongly encouraged generally. They should be provided in Excel format, one file for each relevant figure, with the linked figure noted directly in the file. They should be clearly labelled such that individual experiments and/or animals are labelled (for example, across a time course if applicable). For imaging source data, we encourage deposition to a relevant repository, such as figshare (<https://figshare.com/>) or the Image Data Resource (<https://idr.openmicroscopy.org>).

- that unprocessed scans are clearly labelled and match the gels and western blots presented in figures.
- that control panels for gels and western blots are appropriately described as loading on sample processing controls

-- all images in the paper are checked for duplication of panels and for splicing of gel lanes.

Please include any references for the Methods at the end of the reference list. Any citations in the Supplemental Information will need inclusion in a separate SI reference list.

It is a condition of publication that you include a statement before the acknowledgements naming the author to whom correspondence and requests for materials should be addressed.

Finally, we require authors to include a statement of their individual contributions to the paper -- such as experimental work, project planning, data analysis, etc. -- immediately after the acknowledgements. The statement should be short, and refer to authors by their initials. For details please see the Authorship section of our joint Editorial policies at http://www.nature.com/authors/editorial_policies/authorship.html

We will not send your revised paper for further review if, in the editors' judgement, the referees' comments on the present version have been addressed. If the revised paper is in Nature Microbiology format, in accessible style and of appropriate length, we shall accept it for publication immediately.

Please resubmit electronically

- * the final version of the text (not including the figures) in either Word or Latex.
- * publication-quality figures. For more details, please refer to our Figure Guidelines, which is available here: <https://www.nature.com/documents/NRJs-guide-to-preparing-final-artwork.pdf>
- * Extended Data & Supplementary Information, as instructed
- * a point-by-point response to any issues raised by our referees and to any editorial suggestions.
- * any suggestions for cover illustrations, which should be provided at high resolution as electronic files. Please note that such pictures should be selected more for their aesthetic appeal than for their scientific content. I am sure you will understand that we cannot make any promise as to whether any of your suggestions might be selected for the cover of Nature Microbiology.

Please use the following link to access your home page:

{REDACTED}

* This url links to your confidential homepage and associated information about manuscripts you may have submitted or be reviewing for us. If you wish to forward this e-mail to co-authors, please delete this link to your homepage first.

Please also send the following forms as a PDF by email to microbiology@nature.com.

* Please sign and return the attached form.

* Or, if the corresponding author is either a Crown government employee (including Great Britain and Northern Ireland, Canada and Australia), or a US Government employee, please sign and return the [Licence to Publish form for Crown government employees](http://www.nature.com/documents/snl-ltp-crown.docx), or a [Licence to Publish form for US government employees](http://www.nature.com/documents/snl-ltp-govus.docx).

* Should your Article contain any items (figures, tables, images, videos or text boxes) that are the same as (or are adaptations of) items that have previously been published elsewhere and/or are owned by a third party, please note that it is your responsibility to obtain the right to use such items and to give proper attribution to the copyright holder. This includes pictures taken by professional photographers and images downloaded from the internet. If you do not hold the copyright for any such item (in whole or part) that is included in your paper, please complete and return this [Third Party Rights Table](http://www.nature.com/documents/thirdpartyrights-origres.doc), and attach any grant of rights that you have collected.

For more information on our licence policy, please consult <http://npg.nature.com/authors>.

AUTHORSHIP

CONSORTIA -- For papers containing one or more consortia, all members of the consortium who contributed to the paper must be listed in the paper (i.e., print/online PDF). If necessary, individual authors can be listed in both the main author list and as a member of a consortium listed at the end of the paper. When submitting your revised manuscript via the online submission system, the consortium name should be entered as an author, together with the contact details of a nominated consortium representative. See <https://www.nature.com/authors/policies/authorship.html> for our authorship policy and <https://www.nature.com/documents/nr-consortia-formatting.pdf> for further consortia formatting guidelines, which should be adhered to prior to acceptance.

ORCID

Nature Microbiology is committed to improving transparency in authorship. As part of our efforts in this direction, we are now requesting that all authors identified as 'corresponding author' create and link their Open Researcher and Contributor Identifier (ORCID) with their account on the Manuscript Tracking System (MTS) prior to acceptance. ORCID helps the scientific community achieve unambiguous attribution of all scholarly contributions. For more information please visit <http://www.springernature.com/orcid>

For all corresponding authors listed on the manuscript, please follow the instructions in the link below to link your ORCID to your account on our MTS before submitting the final version of the manuscript. If you do not yet have an ORCID you will be able to create one in minutes.
<https://www.springernature.com/gp/researchers/orcid/orcid-for-nature-research>

IMPORTANT: All authors identified as 'corresponding author' on the manuscript must follow these instructions. Non-corresponding authors do not have to link their ORCIDs but are encouraged to do so. Please note that it will not be possible to add/modify ORCIDs at proof. Thus, if they wish to have their ORCID added to the paper they must also follow the above procedure prior to acceptance.

To support ORCID's aims, we only allow a single ORCID identifier to be attached to one account. If you have any issues attaching an ORCID identifier to your MTS account, please contact the [Platform Support Helpdesk](http://platformsupport.nature.com/).

Nature Research journals [encourage authors to share their step-by-step experimental protocols](https://www.nature.com/nature-research/editorial-policies/reporting-standards#protocols) on a protocol sharing platform of their choice. Nature Research's Protocol Exchange is a free-to-use and open resource for protocols; protocols deposited in Protocol Exchange are citable and can be linked from the published article. More details can be found at www.nature.com/protocolexchange/about.

We hope to hear from you within two weeks; please let us know if the revision process is likely to take longer.

Reviewer Comments:

Reviewer #1 (Remarks to the Author):

This is still a delightful paper to read, and the authors have done a wonderful job addressing all of the comments from the reviewers.

Reviewer #2 and #3 had no further comments.

Final Decision Letter:

Dear Jill,

I am pleased to accept your Article "Large Freshwater Phages with the Potential to Augment Aerobic Methane Oxidation" for publication in Nature Microbiology. Thank you for having chosen to submit your work to us and many congratulations.

Before your manuscript is typeset, we will edit the text to ensure it is intelligible to our wide readership and conforms to house style. We look particularly carefully at the titles of all papers to ensure that they are relatively brief and understandable.

The subeditor may send you the edited text for your approval. Once your manuscript is typeset you will receive a link to your electronic proof via email within 20 working days, with a request to make any corrections within 48 hours. If you have queries at any point during the production process then please contact the production team at rjsproduction@springernature.com. Once your paper has been scheduled for online publication, the Nature press office will be in touch to confirm the details.

Acceptance of your manuscript is conditional on all authors' agreement with our publication policies (see www.nature.com/nmicrobiolate/authors/gta/content-type/index.html). In particular your manuscript must not be published elsewhere and there must be no announcement of the work to any media outlet until the publication date (the day on which it is uploaded onto our website).

The Author's Accepted Manuscript (the accepted version of the manuscript as submitted by the author) may only be posted 6 months after the paper is published, consistent with our [self-archiving embargo](http://www.nature.com/authors/policies/license.html). Please note that the Author's Accepted Manuscript may not be released under a Creative Commons license. For Nature Research Terms of Reuse of archived manuscripts please see: <http://www.nature.com/authors/policies/license.html#terms>
